# Customized 2D CNN Model for the Automatic Emotion Recognition Based on EEG Signals

Farzad Baradaran [1], Ali Farzan [1,*], Sebelan Danishvar [2,*] and Sobhan Sheykhivand [3]

1   Department of Computer Engineering, Shabestar Branch, Islamic Azad University, Shabestar 53816-37181, Iran
2   College of Engineering, Design and Physical Sciences, Brunel University London, Uxbridge UB8 3PH, UK
3   Department of Biomedical Engineering, University of Bonab, Bonab 55517-61167, Iran;
    s.sheykhivand@tabrizu.ac.ir
*   Correspondence: alifarzan402@gmail.com (A.F.); sebelan.danishvar@brunel.ac.uk (S.D.)

**Abstract:** Automatic emotion recognition from electroencephalogram (EEG) signals can be considered as the main component of brain–computer interface (BCI) systems. In the previous years, many researchers in this direction have presented various algorithms for the automatic classification of emotions from EEG signals, and they have achieved promising results; however, lack of stability, high error, and low accuracy are still considered as the central gaps in this research. For this purpose, obtaining a model with the precondition of stability, high accuracy, and low error is considered essential for the automatic classification of emotions. In this research, a model based on Deep Convolutional Neural Networks (DCNNs) is presented, which can classify three positive, negative, and neutral emotions from EEG signals based on musical stimuli with high reliability. For this purpose, a comprehensive database of EEG signals has been collected while volunteers were listening to positive and negative music in order to stimulate the emotional state. The architecture of the proposed model consists of a combination of six convolutional layers and two fully connected layers. In this research, different feature learning and hand-crafted feature selection/extraction algorithms were investigated and compared with each other in order to classify emotions. The proposed model for the classification of two classes (positive and negative) and three classes (positive, neutral, and negative) of emotions had 98% and 96% accuracy, respectively, which is very promising compared with the results of previous research. In order to evaluate more fully, the proposed model was also investigated in noisy environments; with a wide range of different SNRs, the classification accuracy was still greater than 90%. Due to the high performance of the proposed model, it can be used in brain–computer user environments.

**Keywords:** emotion recognition; deep learning; EEG; music; CNN

## 1. Introduction

In the near future, computers will quickly become a pervasive part of human life. However, they are emotionally blind and cannot understand human emotional states [1]. Reading and understanding human emotional states can maximize human–computer interaction (HCI) performance [2]. Therefore, the exchange of this information and the recognition of the user's affective states are considered necessary to increase human–computer interaction [3]. A person's emotional state can be recognized through physiological signs, such as electroencephalography and electrodermal response, as well as physiological indicators, such as facial signs [4]. However, the diagnosis based on physiological indicators is less often considered and used due to its sensitivity to social coverage. Among physiological signals, EEG is the most popular and widely used signal to detect different emotions [5,6]. Evidence shows that there is a very strong correlation between this signal and emotions, such as happiness, sadness, and anger. Therefore, the use of these non-invasive technologies can enable us to develop an emotion recognition system that can be used in everyday life [7,8].

In general, excitement is a group of physiological reactions produced by the human body under multiple external, physical, and mental stimuli [9]. In emotional models, scientists and psychologists divide human emotions into six main categories: sadness, happiness, surprise, fear, anger, and disgust [10]. Various stimuli, such as events, images [11], music [12], or movies [13], have been used to evoke emotions in previous research. Among these stimuli, music is known as the fastest and most effective stimulus for emotional induction. Determining the underlying truth of emotions is difficult because there is no clear definition of emotions, but the best way to determine or interpret emotions during testing is to subjectively rate emotional trials or report them by the test subject. For example, subjective ratings of emotional tests are widely used by researchers [14]. Self-reports can be collected from volunteers using questionnaires or designed instruments. The Self-Assessment Manikin (SAM) is one of these tools designed to assess people's emotional experiences. SAM is a brief self-report questionnaire that uses images to express the scales of pleasure, arousal, and dominance. Given its non-verbal design, the questionnaire is readable by people regardless of age, language skill, or other educational factors [15].

Many studies have been conducted in order to automatic emotion recognition. In the following, previous works will be examined along with their advantages and disadvantages.

Li et al. [16] presented a new model to automatically recognize emotion from EEG signals. These researchers used 128 channels to record the EEG signal. In addition, they identified active channels in 32 volunteers with Correlation-based Feature Selection (CFS) and reported that 5 channels, namely T3, F3, Fp1, O2, and Fp2, have a great effect on emotion recognition. These researchers used the Genetic Algorithm (GA) to reduce the dimensions of the feature vector and used the t-test to verify the correctness of the selected features. Finally, K-Nearest Neighbors (KNN), Random Forest (RF), Bayesian, and Multilayer Perceptron (MLP) classifiers have been used for classification. Yimin et al. [17] recognized the four emotions of happiness, sadness, surprise, and peace from EEG signals. They used eight volunteers to record the EEG signal. These researchers used four classifiers, namely RF, Linear Discriminant Analysis (LDA), Support Vector Machine (SVM), and C4.5, for the classification part and concluded that the C4.5 classifier had a better performance in detecting emotion. Hassanzadeh et al. [18] used a Fuzzy Parallel Cascade (FPC) model to detect emotion. For their experiment, these researchers used a musical stimulus with 15 volunteers. They also compared their proposed model with Recurrent Neural Networks (RNN). Finally, the Mean Squared Error (MSE) of these researchers for the classification of the two classes of valence and arousal is reported to be approximately 0.089, which is lower compared with other models. Panayo et al. [19] used Deep Neural Networks (DNNs) to recognize emotion from EEG signals. They conducted their experiment on 12 people. Their proposed network architecture consisted of six convolutional layers. In addition, these researchers compared their proposed algorithm with SVM and concluded that CNN had better performance in emotion recognition than comparative methods. Chen et al. [20] used EEG signals to automatically classify two classes of emotion. These researchers used parallel RNNs in their proposed algorithm. The final reported accuracy for valence and arousal class classification based on their proposed algorithm was 93.64% and 93.26%, respectively. He et al. [21] used dual Wavelet Transform (WT) to extract features from EEG signals in order to recognize emotion. In addition, these researchers, after feature extraction, used recursive units to train their model. Finally, they achieved an accuracy of 85%, 84%, and 87% for positive, negative, and neutral emotion classes, respectively. Sheykhivand et al. [22] used 12 channels of EEG signals for the automatic recognition of emotion. For this purpose, these researchers used a combination of RNN and CNN for feature selection/extraction and classification. In their proposed model, they identified three different states of emotion, including positive, negative, and neutral, using musical stimulation and achieved 96% accuracy. Among the advantages of their model, the classification accuracy was high, but the computational complexity can be considered as the disadvantage of this research. Er et al. [23] presented a new model for automatic emotion recognition from EEG signals. These researchers used transfer-learning networks, such as VGG16 and AlexNet,

in their proposed model. They achieved satisfactory results based on the VGG16 network in order to classify four different basic emotional classes, including happy, relaxed, sad, and angry. Among the advantages of this research, low computational complexity and low classification accuracy can be considered as disadvantages of this research. Zhao et al. [24] presented a new model for automatic emotion recognition. Their model consisted of two different parts. The first part consisted of a novel multi-feature fusion network that used spatiotemporal neural network structures to learn spatiotemporal distinct emotional information for emotion recognition. In this network, two common types of features, time domain features (differential entropy and sample entropy) and frequency domain features (power spectral density), were extracted. Then, in the second part, they were classified into different classes by Softmax and SVM. These researchers used the DEAP dataset to evaluate their proposed model and achieved promising results. However, computational complexity can be considered a disadvantage of this research. Nandini et al. [25] used multi-domain feature extraction and different time–frequency domain techniques and wavelet-based atomic function to automatically detect emotions from EEG signals. These researchers have used the DEAP database to evaluate their algorithm. In addition, they used machine learning algorithms, such as Random Forest to classify the data and achieved an average accuracy of 98%. Among the advantages of this research is the high classification accuracy. Niu et al. [26] used a two-way deep residual neural network to classify discrete emotions. At first, these researchers divided the EEG signal into five different frequency bands using WT to enter the proposed network. In order to evaluate their algorithm, they collected a dedicated database from seven participants. The classification accuracy reported by these researchers was 94%. Among the problems of this research was the high computational load. Vergan et al. [27] have used deep learning networks to classify three and four emotional classes. These researchers used the CNN network to select and extract features from EEG signals. In order to reduce the deep feature vector, the semi-supervised dimensionality reduction method was used by these researchers. They used two databases, DEAP and SEED, in order to evaluate their proposed method and achieved a high accuracy of 90%. Hu et al. [28] used Feature Pyramid Network (FPN) to improve emotion recognition performance based on EEG signals. In their proposed model, the Differential Entropy (DE) of each recorded EEG channel was extracted as the main feature. These researchers used SVM to score each class. The accuracy reported by these researchers in order to detect the dimension of valence and arousal for the DEAP database was reported as 94% and 96%, respectively. Among the advantages of this research is high classification accuracy. In addition, due to the computational complexity, their proposed model could not be implemented on real-time systems, which can be considered a disadvantage of this research.

As reviewed and discussed, many studies have been conducted and organized for automatic emotion recognition from EEG signals. However, these studies have limitations and challenges. Most previous research has used manual and engineering features in feature extraction/selection. Using manual features requires prior knowledge of the subject and may not be optimal for another subject. In simpler terms, the use of engineering features will not guarantee the optimality of the extracted feature vector. The next limitation of previous research can be considered the absence of a comprehensive and up-to-date database. Existing databases to emotion recognition are limited and are organized based on visual stimulation and, thus, are not suitable for use in deep learning networks. It can almost be said that there is no general and standard database based on auditory stimulation. Many studies have used deep learning networks to detect emotions and have achieved satisfactory results. However, due to computational complexity, these studies cannot be implemented in real-time systems. Accordingly, this research tries to overcome the mentioned limitations and present a new model with high reliability and low computational complexity in order to achieve automatic emotion recognition. To this end, a comprehensive database for emotion recognition based on musical stimuli has been collected in the BCI laboratory of Tabriz University based on EEG signals in compliance with the necessary standards. The proposed model is based on deep learning, which can

identify the optimal features from the alpha, beta, and gamma bands extracted from the recorded EEG signal to hierarchically and end-to-end classify the basic emotions in two different scenarios. The contribution of this study is organized as follows:

- Collecting a comprehensive database of emotion recognition using musical stimulation based on EEG signals.
- Presenting an intelligent model based on deep learning in order to separate two and three basic emotional classes.
- Using end-to-end deep neural networks, which has led to the elimination of feature selection/extraction block diagram.
- Providing an algorithm based on deep convolutional networks that can be resistant to environmental noise to an acceptable extent.
- Presenting an automatic model that can classify two and three emotional classes with the highest accuracy and the least error compared with previous research.

The remainder of the article is written and organized as follows: Section 2 is related to materials and methods, and in this section, the method of data collection and the mathematical background related to deep learning networks are described. Section 3 is related to the proposed model, which describes the data preprocessing and the proposed architecture. Section 4 presents the simulation results and compares the obtained results with previous research. Section 5 discusses the applications related to the current research. Finally, Section 6 addresses the conclusion.

## 2. Materials and Methods

In this section, firstly, the method of collecting the proposed database of EEG signals for emotion recognition is described. Then, the mathematical background related to common signal processing filters and DNNs is presented.

### 2.1. EEG Database Collection

In order to collect the database, EEG signals were used for emotion recognition from 11 volunteers (5 men and 6 women) in the age range of 18 to 32 years. All volunteers present in the experiment were free of any underlying diseases. In addition, they read and signed the written consent form to participate in the experiment. This experiment was approved by the ethics committee of the university with license 1398.12.1 in the BCI laboratory at Tabriz University. The volunteers were asked to avoid alcohol, medicine, caffeine, and energy drinks 48 h before the test. In addition, volunteers were asked to take a bath the day before the test and avoid using hair-softening shampoos. All the tests were performed at 9 am so that people would have enough energy. Before the experiment, Beck's popular Depression Mood Questionnaire [29] was used to exclude from the experiment volunteers who suffered from depression. After this test, the candidates who received a score higher than 21, were deemed to have depression disorder and were excluded from the test process. The reason for using this test is that depressive disorder causes a lack of emotional induction in people. In addition, the SAM assessment test in paper form with 9 grades was used to control the dimension of valence and arousal [15]. In the relevant test, a score lower than 3 and a score higher than 6 are considered low grade and high grades, respectively.

In order to record EEG signals, a 21-channel Medicom device according to the 10–20 standard was used. Medicam is a Russian device for recording brain signals, which is widely used in medical clinics and research centers due to its high performance. Silver chloride electrodes, which were organized in the form of a hat, were used in this work. Two electrodes, A1 and A2, were used to reference brain signals. Thus, out of 21 channels, 19 channels are actually available. To avoid EOG signal artifacts, volunteers were asked to keep their eyes closed during EEG signal recording. The sampling frequency was 250 Hz and an impedance matching of 10 k$\Omega$ was used on the electrodes. The recording mode of the signals was also set as bipolar.

Table 1 shows the details of the signal recording of the volunteers in the experiment and the reason for removing some volunteers from the experiment process. To clarify the reason for the exclusion of the volunteers in the experiment, Volunteer 6 was excluded from the experiment due to the low level of positive emotional arousal (<6). In addition, Volunteer 2 was excluded from the continuation of the signal recording process due to depression disorder (21 > 22).

In order to arouse positive and negative emotion in subjects, musical stimulus was used in this study. To this end, 10 pieces of music with happy and sad styles were played for the volunteers through headphones. Each piece of music was played for the volunteers for 1 min, and the EEG signals of the subjects were recorded. Between each music played, the volunteers were given 15 s of rest (neutral) in order to prevent the transfer of produced excitement. An example of the recorded EEG signal for 3 different emotional states is shown in Figure 1.

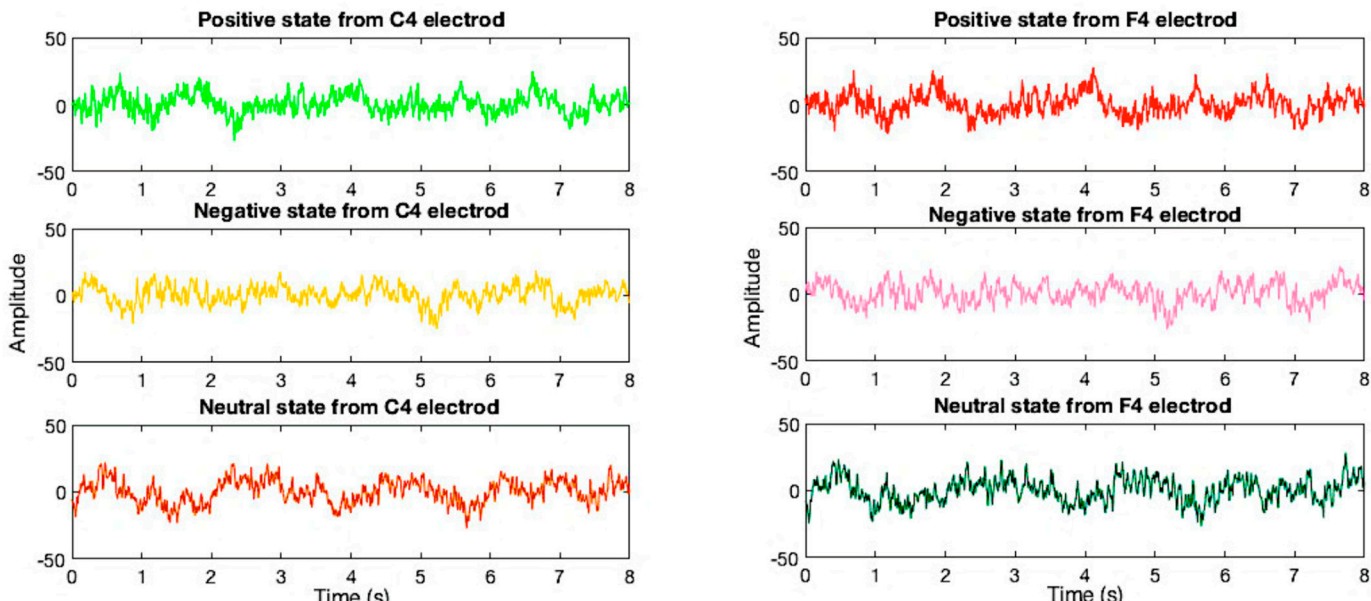

**Figure 1.** An example of EEG signal recorded from C4 and F4 channels for positive, negative, and neutral emotions in Subject 1.

In this way, the signals recorded from subjects for happy songs, sad songs, and relaxation state are labeled as positive emotion, negative emotion, and neutral emotion, respectively. Table 2 shows the Persian songs played for subjects. Figure 2 shows the order of playing music for subjects.

**Table 1.** Details related to recording the signal of volunteers in the experiment.

| Subject | 1 | 2 | 3 | 4 | 5 | 6 | 7 | 8 | 9 | 10 | 11 | 12 | 13 | 14 | 15 | 16 |
|---|---|---|---|---|---|---|---|---|---|---|---|---|---|---|---|---|
| Sex | M | M | F | M | M | M | M | M | M | F | F | F | F | M | F | M |
| Age | 25 | 24 | 27 | 24 | 32 | 18 | 25 | 29 | 30 | 19 | 18 | 20 | 22 | 24 | 23 | 28 |
| BDI | 16 | 22 | 19 | 4 | 0 | 11 | 13 | 19 | 20 | 14 | 22 | 12 | 0 | 12 | 1 | 9 |
| Valence for P emotion | 9 | 6.8 | 6.2 | 7.4 | 5.8 | 5.6 | 7.2 | 7.8 | 7.4 | 6.8 | 7.8 | 8.6 | 6 | 8 | - | 7.4 |
| Arousal for P emotion | 9 | 6.2 | 7.4 | 7.6 | 5 | 5.4 | 7.4 | 7.4 | 7 | 6.6 | 8 | 8.6 | 6 | 8 | - | 8 |
| Valence for N emotion | 2 | 3.6 | 4.2 | 2.4 | 4.4 | 2 | 3.8 | 2.8 | 3.4 | 3.8 | 4.5 | 2 | 2 | 1.8 | - | 1.8 |
| Arousal for N emotion | 1 | 2 | 4.6 | 2.6 | 5.6 | 1.6 | 3.8 | 3 | 5.4 | 3.2 | 3 | 1.2 | 1.2 | 1.6 | - | 2 |
| Result of Test | ACC | REJ | REJ | ACC | REJ | REJ | REJ | ACC | REJ | REJ | REJ | ACC | ACC | ACC | REJ | ACC |
| Reason for rejection | - | Depressed 21 < 22 | Failure in the SAM test | - | Failure in the SAM test | Failure in the P emotion | Failure in the N emotion | - | Failure in the N emotion | Failure in the N emotion | Depressed 21 < 22 | - | - | - | Motion noise | - |

**Table 2.** The music used in the experiment.

| Emotion Sign and Music Number | The Type of Emotion Created in the Subject | The Name of the Music |
|---|---|---|
| N1 | Negative | Advance Income of Isfahan |
| P1 | Positive | Azari 6/8 |
| N2 | Negative | Advance Income of Homayoun |
| P2 | Positive | Azari 6/8 |
| P3 | Positive | Bandari 6/8 |
| N3 | Negative | Afshari piece |
| N4 | Negative | Advance Income of Isfahan |
| P4 | Positive | Persian 6/8 |
| N5 | Negative | Advance Income of Dashti |
| P5 | Positive | Bandari 6/8 |

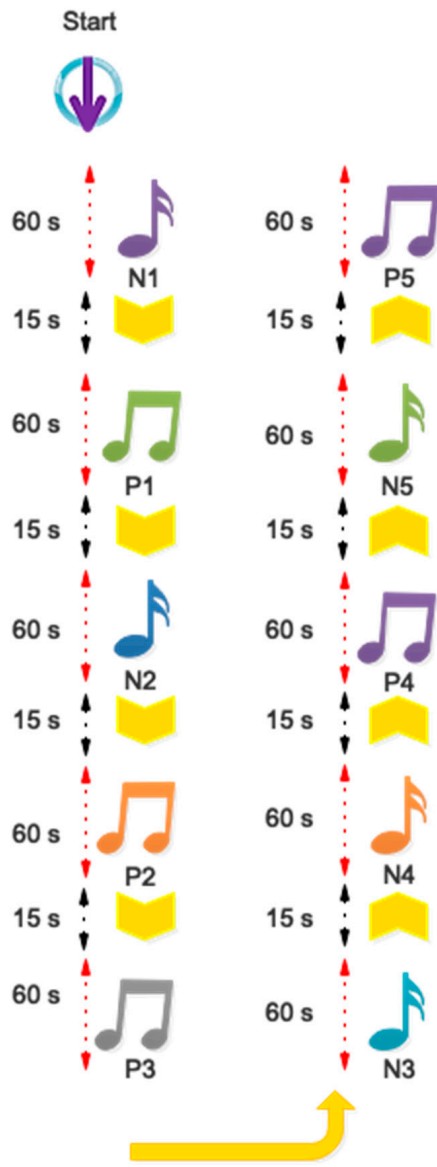

**Figure 2.** Schedule of music played for the volunteers.

*2.2. Signal Processing Filters*

In the field of signal processing, many filtering algorithms are used for signal pre-processing in order to remove motion and environmental noises and to reach the desired frequency range. Among the most popular filters, we mention Notch [30] and Butterworth filters [31], which are also used in this study. In the following section, the mathematical details of each of these filters are examined.

2.2.1. Notch Filter

A Notch filter is a type of band-stop filter, which is a filter that attenuates frequencies within a specific range while passing all other frequencies unaltered. For a Notch filter, this range of frequencies is very narrow. The range of frequencies that a band-stop filter attenuates is called the stopband. The narrow stopband in a Notch filter makes the frequency response resemble a deep notch, which gives the filter its name. It also means that Notch filters have a high Q factor, which is the ratio of center frequency to bandwidth. Notch filters are used to remove a single frequency or a narrow band of frequencies. In audio systems, a Notch filter can be used to remove interfering frequencies, such as power line hum. Notch filters can also be used to remove a specific interfering frequency in radio receivers and software-defined radio. The main application of the notch filter can be considered to remove the frequency of 50 or 60 Hz of city electricity [30].

2.2.2. Butterworth Filter

Among the very popular filters for removing the frequency range above 100 Hz of brain signals is the low-pass Butterworth filter [31]. The Butterworth filter is a type of signal processing filter designed to have a frequency response that is as flat as possible in the passband. It is also referred to as a maximally flat magnitude filter. One of the most important features of this filter is the existence of a flat maximum frequency response in the pass region and no ripple. In addition, its graph tends to a very good approximation with a negative slope to negative infinity.

Butterworth showed that a low-pass filter could be designed whose cutoff frequency was normalized to 1 radian per second; its frequency response can be defined by Equation (1):

$$G(\omega) = \frac{1}{\sqrt{1 + \omega^{2n}}} \tag{1}$$

where $\omega$ is the angular frequency in radians per second and n is the number of poles in the filter—equal to the number of reactive elements in a passive filter. If $\omega = 1$, the amplitude response of this type of filter in the passband is $1/\sqrt{2} \approx 0.7071$, which is half power or $-3$ dB [31].

*2.3. Brief Description of Convolutional Neural Networks Model*

Deep learning can be considered as a subset of Machine Learning (ML), which has been widely used in the previous years in all subjects, including medicine, agriculture, industries, and engineering. CNNs can be considered as the main part of deep learning networks. CNNs have been shown to be a highly successful replacement for traditional neural networks in the development of machine learning classification algorithms. CNN learns in two stages: feed forward and reverse propagation. In general, DCNN is composed of three major layers: convolutional, pooling, and connected layers [32]. The output of a convolutional layer is referred to as feature mapping. The max-pooling layer is typically employed after each convolutional layer and chooses just the maximum values in each feature map. A dropout layer is employed to prevent overfitting; hence, each neuron is thrown out of the network at each stage of training with a probability. A Batch Normalization (BN) layer is commonly used to normalise data within a network and

expedites network training. BN is applied to the neurons' output just before applying the activation function. Usually, a neuron without BN is computed as follows:

$$z = g(w, x) + b; \quad a = f(z) \tag{2}$$

where $g()$ is the linear transformation of the neuron, $w$ is the weight of the neuron, $b$ is the bias of the neurons, and $f()$ is the activation function. The model learns the parameters $w$ and $b$. By adding the BN, the equation can be defined as follows:

$$z = g(w, x); \quad z^N = \left(\frac{z - m_z}{s_z}\right) \cdot \gamma + \beta; \quad a = f(z^N) \tag{3}$$

where $z^N$ is the output of BN, $m_z$ is the mean of the neurons' output, $S_z$ is the standard deviation of the output of the neurons, and $\gamma$ and $\beta$ are learning parameters of BN [33].

One of the most significant components of deep learning is the performance of activation functions because activation functions play an important part in the learning process. An activation function is used after each convolutional layer. Various activation functions, such as ReLU, Leaky-ReLU, ELU, and Softmax, are available to increase learning performance on DNN networks. Since the discovery of the ReLU activation function, which is presently the most often used activation unit, DNNs have come a long way. The ReLU activation function overcomes the gradient removal problem while simultaneously boosting learning performance. The ReLU activation function is described as follows [32]:

$$q(f) = \begin{cases} f & if \ f > 0 \\ 0 & \text{otherwise} \end{cases} \tag{4}$$

The Softmax activation function is defined as follows [26]:

$$\sigma(d)_n = \frac{e^{d_n}}{\sum_{j=1}^{k} s^{d_j}} \ \text{for} \ n = 1, \dots k, \ d = (d_1, \dots, d_k) \in R^k \tag{5}$$

where d is the input vector, the output values $\sigma(d)$ are between 0 and 1, and their sum is equal to 1.

In the prediction step of deep models, a loss function is utilized to learn the error ratio. In machine learning approaches, the loss function is a method of evaluating and describing model efficiency. An optimization approach is then used to minimize the error criterion. Indeed, the results of optimization are used to update hyper parameters [33].

## 3. Proposed Deep Model

In this section, the proposed deep model, which includes pre-processing of recorded data, deep network architecture design, parameter optimization, and training and evaluation sets, is fully described. Figure 3 shows the main framework of the proposed deep model for automatic emotion recognition from EEG signals based on musical stimulation.

### 3.1. Data Pre-Processing

In this section, the method of preprocessing the recorded data, which includes filtering and segmentation, is examined. Brain signals are strongly affected by noise, and it is necessary to remove environmental and motion noises using different filtering algorithms. For this purpose, two filtering algorithms were used in this work to remove artifacts. First, a Notch filter was applied to the recorded EEG signals in order to eliminate the frequency of 50 Hz. Then, considering that emotional arousal occurs only between the ranges of 0.5 to 45 Hz [22,34], a first-order Butterworth filter with a frequency of 0.5 to 45 Hz was used on the EEG data.

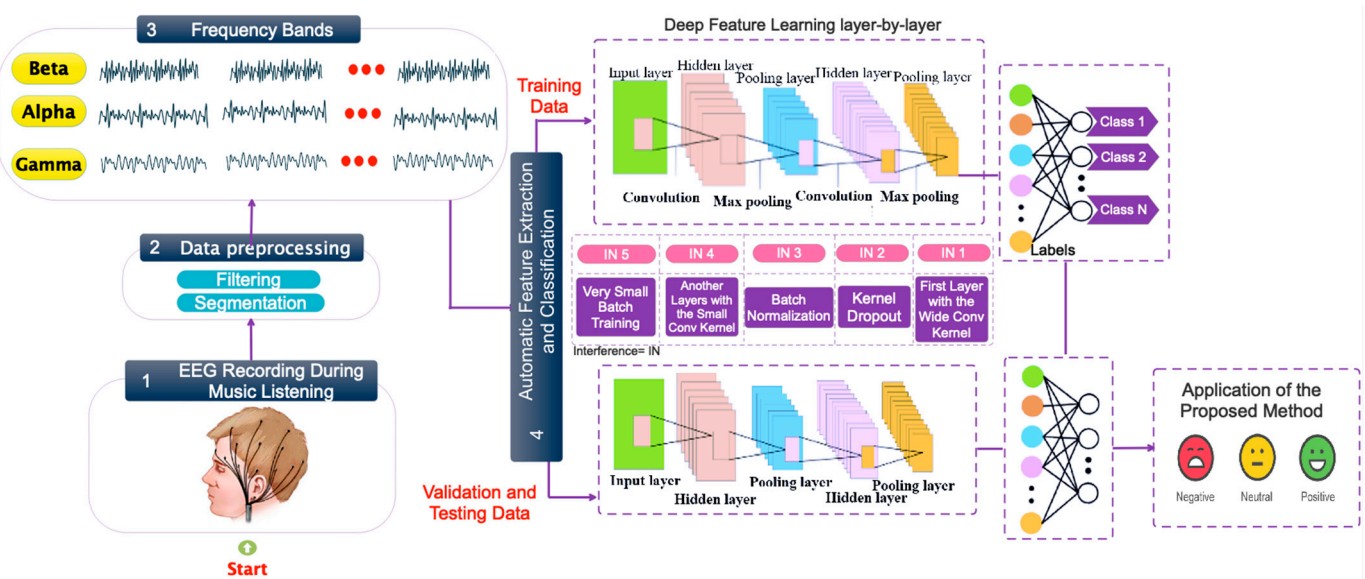

**Figure 3.** The main framework of the proposed deep model.

As is clear, the use of all EEG channels will increase the computational load due to the increase in the dimensions of the feature matrix. Therefore, it is necessary to identify active channels. Emotion-related EEG electrodes were distributed mainly in the prefrontal lobe, temporal lobe margin, and posterior occipital lobe [35]. These regions are precisely in line with the physiological principle of emotion generation. By selecting the electrode distribution, the extracted feature dimension can be greatly reduced. The complexity of calculation can be diminished, and the experiment is more straightforward and easier to carry out. Based on this, only the electrodes of the prefrontal lobe, temporal lobe margin, and posterior occipital lobe, which include Pz, T3, C3, C4, T4, F7, F3, Fz, F4, F8, Fp1, and Fp2, were used for processing [35].

We selected 5 min (30 s × 5 min = 300 s) of the signals recorded from electrodes for each positive and negative class. Considering that the sampling frequency was 250 Hz, we had 75,000 available samples for each class. In the next step, 3 frequency bands $\alpha$, $\beta$, and $\gamma$ were extracted from the data using 8th Daubechies WT [36]. For the first subject and the positive emotional state, these frequency bands for one segment are presented in Figure 4. Then, in order to avoid the phenomenon of overfitting, overlapping operations were performed on the data obtained from the selected electrodes according to Figure 5. According to this operation for the two-class scenario, the recorded data from the selected electrodes was divided into 540 samples of 8 s each. All the mentioned steps were repeated for the second scenario with the difference in data length. Finally, in this research, the input data for both scenarios were applied to the proposed network in the form of images. Thus, the input data for the first and second scenarios were equal to $(7560) \times (36 \times 2000 \times 1)$ and $(11340) \times (36 \times 2000 \times 1)$, respectively. The input images from the extracted frequency bands are shown in Figure 6.

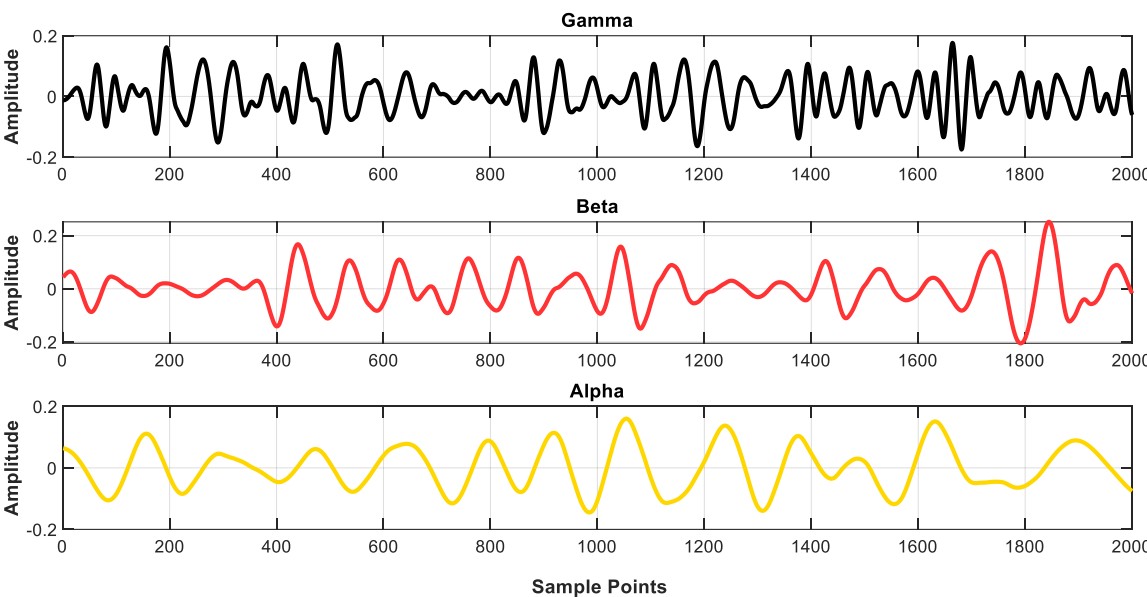

**Figure 4.** Extracted frequency bands for a segment for the positive emotional state of the first subject.

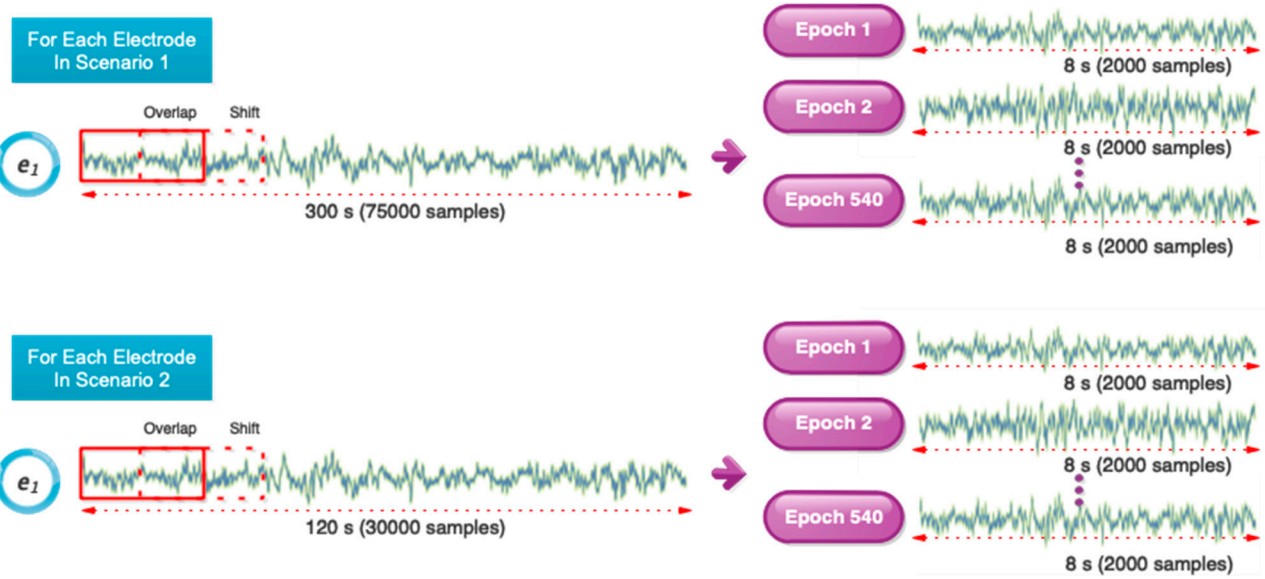

**Figure 5.** Overlap operation performed on the EEG signal for each electrode in positive, negative, and neutral emotion.

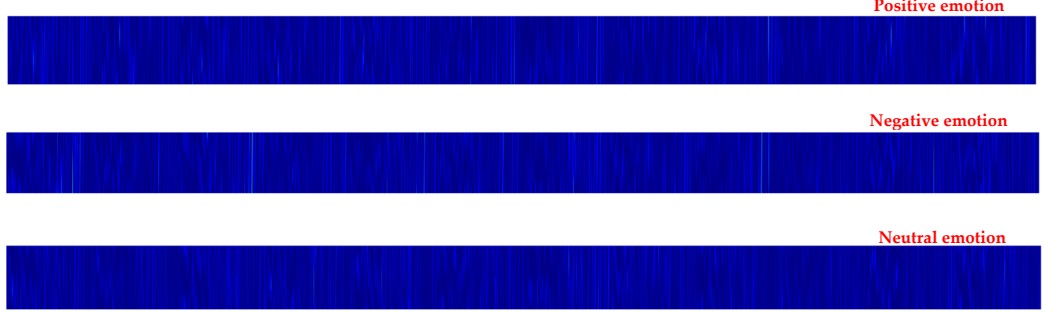

**Figure 6.** Input formed images for positive, negative, and neutral emotion for the first subject based on extracted $\alpha$, $\beta$, and $\gamma$ frequency bands.

### 3.2. Deep Architectural Details

For the proposed deep network architecture, a combination of six convolutional layers along with two fully connected layers was used. The order of the layers was as follows:

I.  One drop-out layer.
II.  A 2D Convolution layer with the Leaky-ReLu nonlinear function and a Max-Pooling layer with Batch Normalization are added.
III.  The architecture of the previous stage is repeated three more times.
IV.  A Convolution 2D layer is added with the Leaky-ReLu nonlinear function along with the Batch Normalization.
V.  The architecture of the previous stage is repeated one more time.
VI.  The output of the previous architecture is connected to the two Fully Connected layers, which are used in the last layer of the Softmax function to access the outputs and emotion recognition.

The graphic representation of the mentioned proposed architecture is shown in Figure 7.

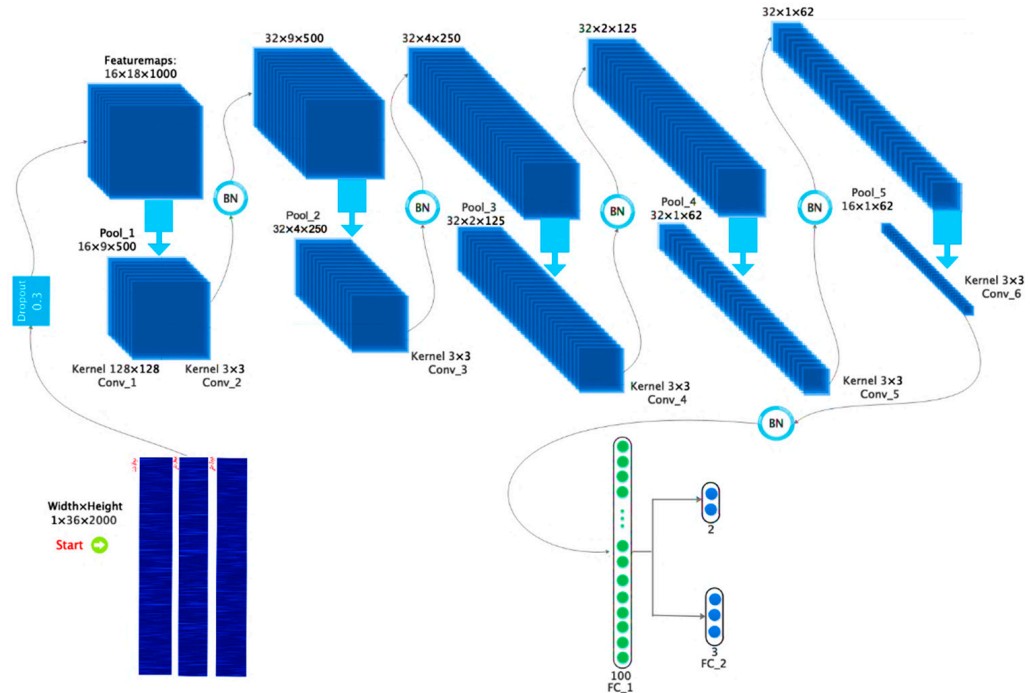

**Figure 7.** Graphical details of the designed network architecture along with the sizes of filters, layers, etc.

The hyper-parameters related to the proposed model were carefully adjusted in order to achieve the best efficiency and convergence by the trial-and-error method. Accordingly, the Cross-Entropy objective function and RmSprop optimization with a learning rate of 0.001 and a batch size of 10 were selected. More details related to the size of the filters, the number of steps, and the type of layers used are shown in Table 3.

In the design of the proposed architecture, we attempted to take into account the best dimensions for the size and strides of filters, optimizers, etc., so that the proposed model could perform best in terms of different evaluation criteria in order to emotion recognition. Table 4 shows the number of different layers of the network, different types of optimizers, sizes of filters and steps, etc., which were used in choosing the optimal mode of the proposed architecture. According to Table 4, the best possible state of the used parameters was selected in the architecture of the proposed model.

**Table 3.** The details of the network architecture, including the size of the filters, the number of layers, and the type of layers.

| Padding | Number of Filters | Strides | Size of Kernel and Pooling | Output Shape | Activation Function | Layer Type | L |
|---|---|---|---|---|---|---|---|
| **Yes** | 16 | 2 | 128 × 128 | (None, 18, 1000, 16) | Leaky ReLU | Convolution 2-D | 0–1 |
| **No** | - | 2 | 2 × 2 | (None, 9, 500, 16) | - | Max-Pooling 2-D | 1–2 |
| **Yes** | 32 | 1 | 3 × 3 | (None, 9, 500, 32) | Leaky ReLU | Convolution 2-D | 2–3 |
| **No** | - | 2 | 2 × 2 | (None, 4, 250, 32) | - | Max-Pooling 2-D | 3–4 |
| **Yes** | 32 | 1 | 3 × 3 | (None, 4, 250, 32) | Leaky ReLU | Convolution 2-D | 4–5 |
| **No** | - | 2 | 2 × 2 | (None, 2, 125, 32) | - | Max-Pooling 2-D | 5–6 |
| **Yes** | 32 | 1 | 3 × 3 | (None, 2, 125, 32) | Leaky ReLU | Convolution 2-D | 6–7 |
| **No** | - | 2 | 2 × 2 | (None, 1, 62, 32) | - | Max-Pooling 2-D | 7–8 |
| **Yes** | 32 | 1 | 3 × 3 | (None, 1, 62, 32) | Leaky ReLU | Convolution 2-D | 8–9 |
| **Yes** | 16 | 1 | 3 × 3 | (None, 1, 62, 16) | Leaky ReLU | Convolution 2-D | 10–11 |
| - | - | - | - | (None, 100) | Leaky ReLU | FC | 11–12 |
| - | - | - | - | (None, 2–3) | Softmax | FC | 12–13 |

**Table 4.** The details of the designed network architecture, including the size of the filters, the number of layers, and the type of layers.

| | Parameters | Search Space | Optimal Value |
|---|---|---|---|
| | Optimizer | RMSProp, Adam, Sgd, Adamax, and Adadelta | RMSProp |
| | Cost function | MSE, Cross-entropy | Cross-Entropy |
| Number of | Convolution layers | 3, 5, 6, 11, 15 | 6 |
| | Filters in the first convolution layer | 16, 32, 64, 128 | 16 |
| | Filters in the second convolution layer | 16, 32, 64, 128 | 32 |
| | Filters in other convolution layers | 16, 32, 64, 128 | 32 |
| Size of filter in the | First convolution layer | 3, 16, 32, 64, 128 | 128 |
| | Other convolution layers | 3, 16, 32, 64, 128 | 3 |
| Dropout rate | Before the first convolution layer | 0, 0.2, 0.3, 0.4, 0.5 | 0.3 |
| | After the first convolution layer | 0, 0.2, 0.3, 0.4, 0.5 | 0.3 |
| | Batch size | 4, 8, 10, 16, 32, 64 | 10 |
| | Learning rate | 0.01, 0.001, 0.0001 | 0. 001 |

The number of divided samples for each of the training, test, and validation sets are examined in this section. Based on this, the total number of samples in this study for the first and second scenarios was 7560 and 11,340, respectively, of which 70% were randomly selected for the training set (5292 samples for the two-class state and 7938 samples for the three-class state), 10% of the dataset was selected for the validation set (756 samples for the two class state and 1134 samples for the three class state), and 20% of the dataset (1512 samples for the two class state and 2268 samples for the three class state) was selected for the test set. The collection related to model training and evaluation is shown in Figure 8.

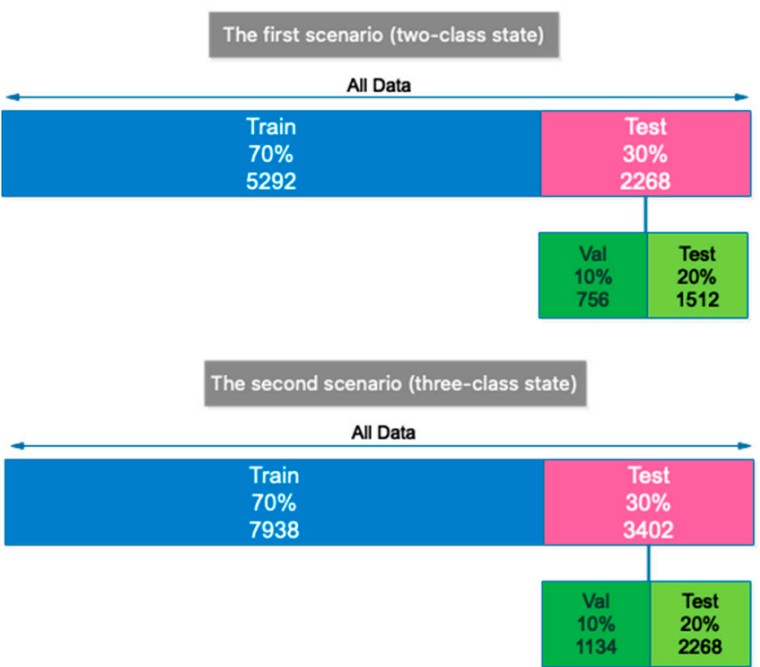

**Figure 8.** Dataset distribution along the data recognition process.

## 4. Experimental Results

Python programming language under Keras and TensorFlow was used to simulate the proposed deep model. All simulation results are extracted from a computer system with 16 GB RAM, a 2.8 GHz CPU, and a 2040 GPU.

Figure 9 depicts the classification error and accuracy graph for various scenarios for training and validation data in 200 network iterations. Figure 9a shows that the network error for the two-class state reached a stable state by increasing the algorithm iteration in the 165th iteration. Figure 9 shows that after 200 repetitions, the proposed method for emotion recognition achieved 98% and 96% accuracy in the two-class and the three-class states, respectively. Figure 10 depicts the confusion matrix used to classify the scenarios under consideration. According to Figure 10, the proposed deep network's performance is very promising. Table 5 also shows the values of accuracy, sensitivity, specificity, and precision for each emotion in different scenarios. As can be seen, all the values obtained for each class of the two considered scenarios were greater than 90%. A visualization of the samples before and after entering the network was considered to demonstrate the more accurate performance of the proposed network. Figure 11 shows a TSen diagram with this visualization. As can be seen, the proposed model successfully separated the samples related to each emotion in each scenario. This positive outcome is due to the proposed improved CNN architecture. Figure 12 depicts the Receiver Operating Characteristic Curve (ROC) analysis for different scenario classifications. An ROC is a graphical plot that illustrates the diagnostic ability of a classifier system as its discrimination threshold is varied. In this diagram, the farther the curve is from the bisector and the closer it is to the vertical line on the left, the better the performance of the classifier. As is well known, each emotion class in the ROC analysis has a score between 0.9 and 1, indicating excellent classification performance. Based on the results, it is possible to conclude that the proposed deep model for classifying different emotional classes was very efficient and met the relevant expectations. However, in order to conduct further analysis, the obtained results must be compared with other studies. The findings will be compared with other previous studies and methods for this purpose.

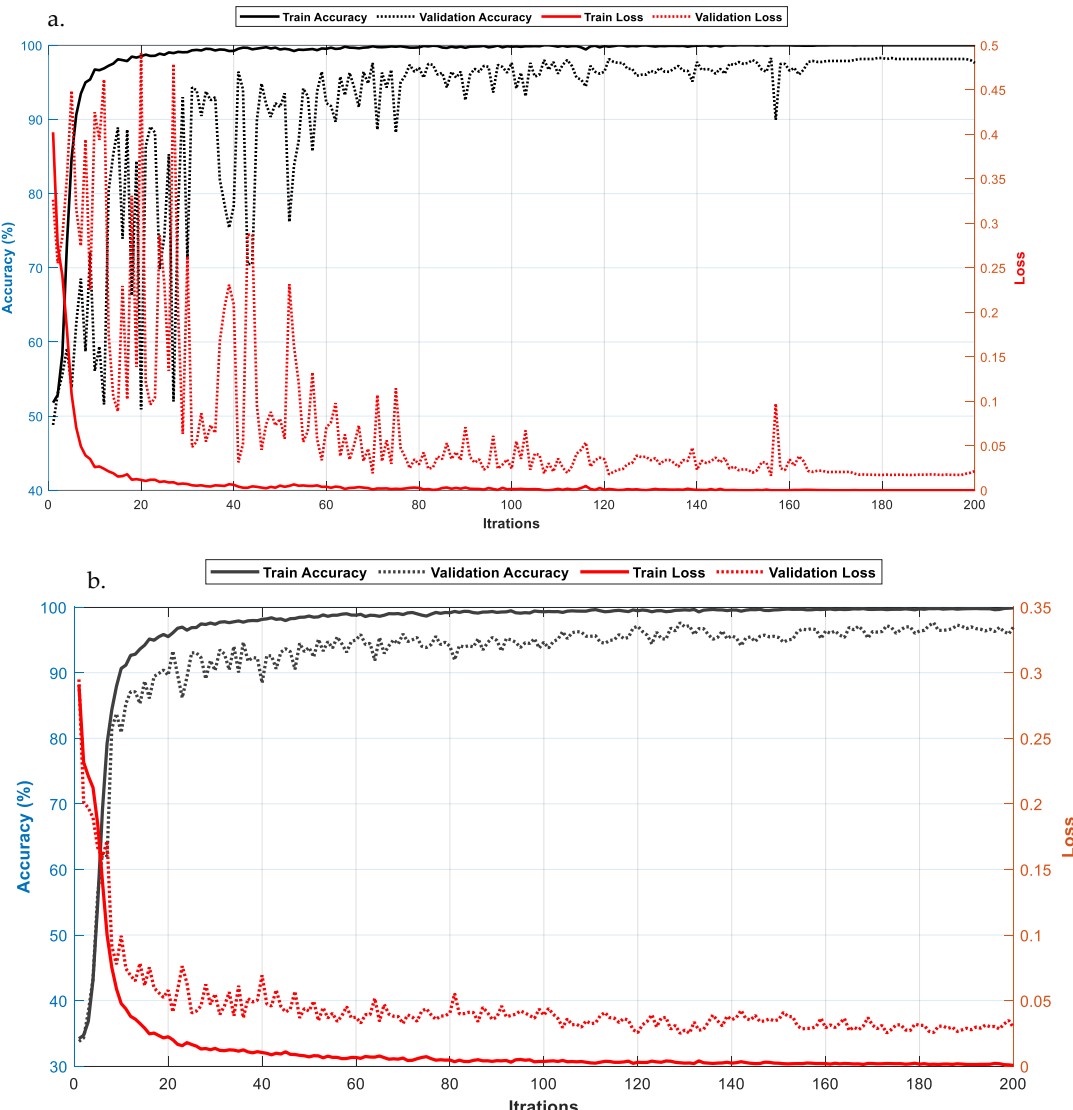

**Figure 9.** The proposed deep model's performance in classifying different scenarios (**a**,**b**) in terms of accuracy and classification error in 200 iterations.

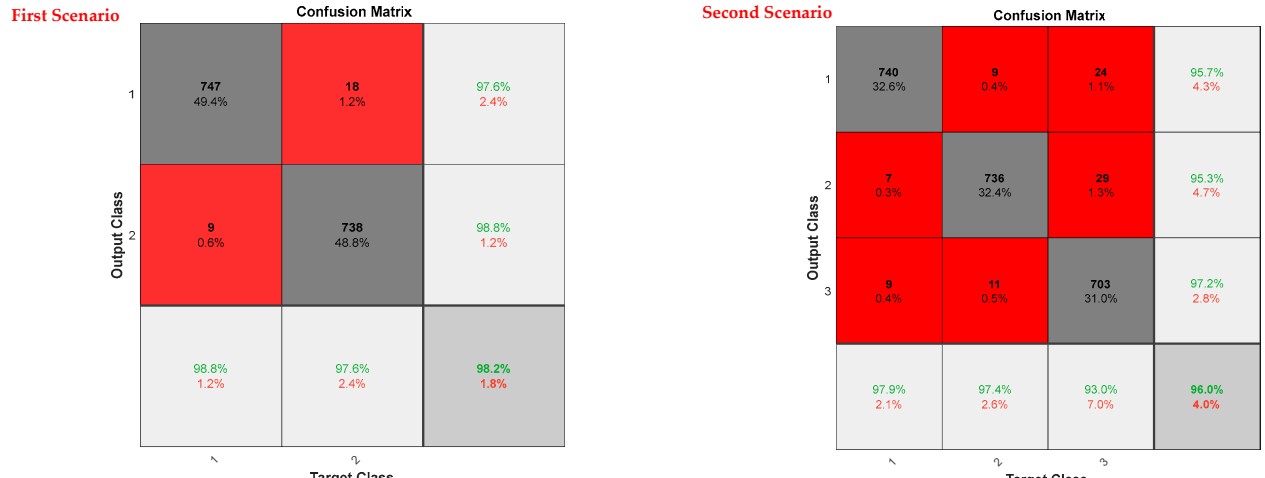

**Figure 10.** Confusion matrix for classifying various scenarios.

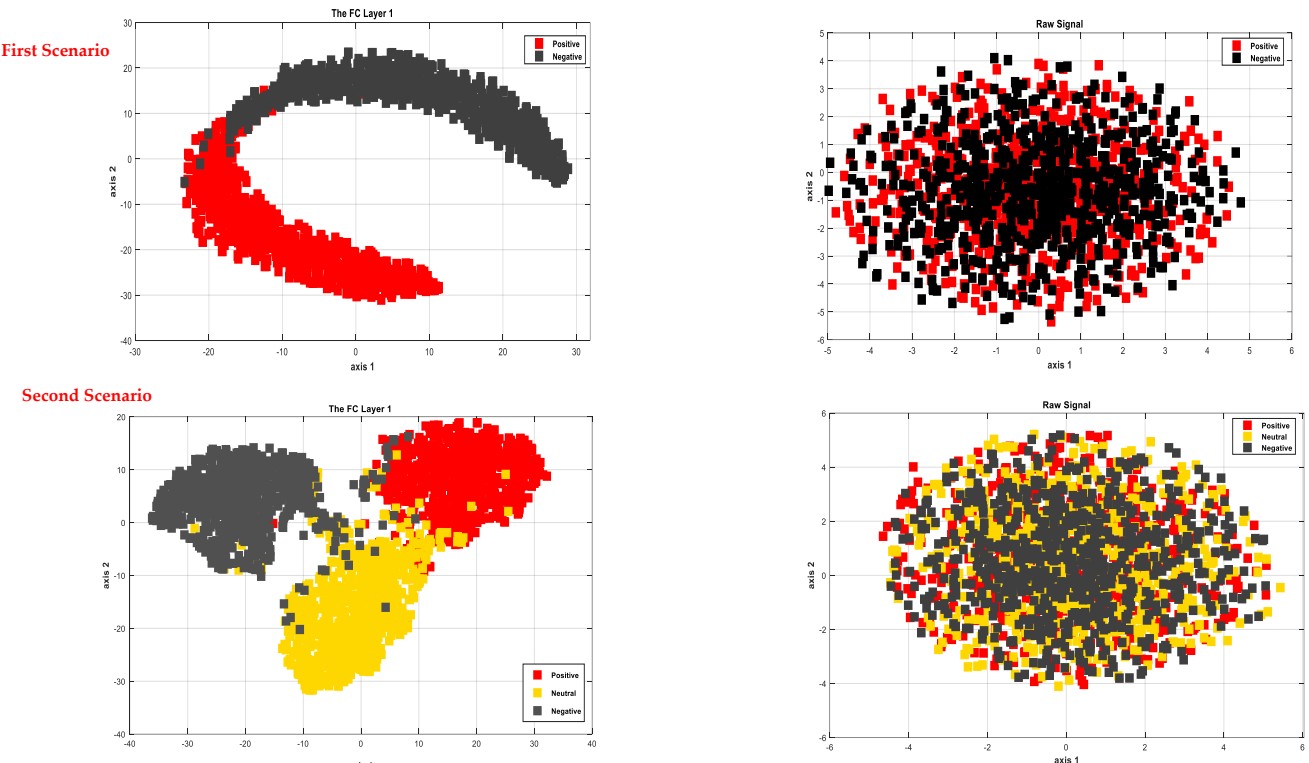

**Figure 11.** Visualization of test samples before and after entering the proposed deep model for different scenarios.

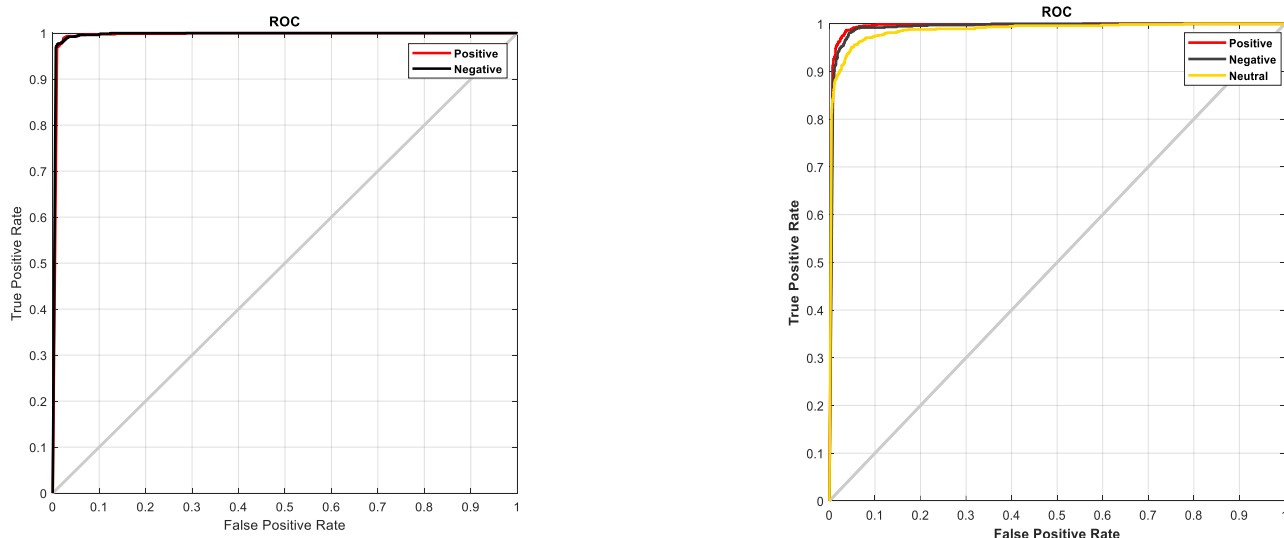

**Figure 12.** ROC analysis to classify different scenarios.

Table 6 compares previous studies, as well as the methods employed in each study, with the proposed improved deep model. As shown in Table 6, the proposed method achieved the highest accuracy when compared with previous works. However, this comparison does not appear to be fair because the databases under consideration are not identical. As a result, to use the proposed recorded database, it is necessary to simulate and evaluate the methods used in prior research.

**Table 5.** The accuracy, sensitivity, specificity, and precision achieved by each class for different scenarios.

| First Scenario (P and N) | Positive | Negative | |
|---|---|---|---|
| Sensitivity | 98.2 | 98.2 | |
| Accuracy | 98.8 | 97.6 | |
| Specificity | 97.6 | 98.8 | |
| Precision | 97.6 | 98.8 | |
| Second Scenario | Positive | Negative | Neutral |
| Sensitivity | 97.8 | 97.5 | 96.7 |
| Accuracy | 95.7 | 95.3 | 97.2 |
| Specificity | 98.9 | 98.6 | 96.5 |
| Precision | 97.9 | 97.4 | 93 |

**Table 6.** Comparing the performance of prior research with the proposed model.

| Study | Stimulus | Methods | Number of Emotions Considered | ACC% |
|---|---|---|---|---|
| Zhao et al. [37] | Music | Deep local domain | 4 | 89 |
| Chanel et al. [38] | Video Games | Frequency bands extraction | 3 | 63 |
| Jirayucharoensak et al. [39] | Video Clip | Principal component analysis | 3 | 49.52 |
| Er et al. [23] | Music | VGG16 | 4 | 74 |
| Sheykhivand et al. [22] | Music | CNN-LSTM | 3 | 96 |
| Hou et al. [28] | Video Clip | FPN+SVM | 4 | 95.50 |
| **Proposed model** | Music | Customized CNN | 3 | **98** |

To more accurately evaluate the proposed model, the deep network architecture presented in this study was compared with other common methods and previous research used for automatic emotion recognition. In this regard, two methods based on raw signal feature learning and engineering feature extraction (manual) were used along with MLP classifiers, CNN-1D, CNN-LSTM (1D), and the proposed CNN-2D model. The gamma band was extracted from the recorded EEG signals for engineering features (using a 5-level Daubechies WT). From the obtained gamma band, two Root Mean Square (RMS) and Standard Deviation (SD) features were extracted. Based on this, the input dimensions for the first and second scenarios were $(2 \times 7 \times 540) \times (e \times 2)$ and $(3 \times 7 \times 540) \times (e \times 2)$, respectively. Following that, MLP, CNN-1D, CNN-LSTM (1D), and proposed CNN-2D networks were used to classify the extracted feature vector. The raw signals were classified using expressed networks for feature learning, with no manual feature extraction or selection. The MLP network had two fully connected layers, the last of which had two neurons (for the two-class state) and three neurons (for the three-class state). Following [22], CNN-1D and CNN-LSTM (1D) network architectures were considered. To improve the performance of the expressed networks, their hyperparameters were adjusted on the basis of the type of data. The results of this comparison are shown in Table 7 and Figure 13. According to Table 7, feature learning from raw signals for CNN-1D, CNN-LSTM (1D), and proposed CNN-2D deep networks were continually improved, and these networks could learn important features layer by layer, resulting in two-class and three-class scenarios with accuracy greater than 90%. On the contrary, as can be seen from the engineering features used as input in CNN-1D, CNN-LSTM (1D), and CNN-2D deep networks, these networks did not improve recognition.

When feature learning and engineering features were compared, feature learning from raw data with CNN-1D, CNN-LSTM (1D), and CNN-2D deep networks outperformed engineering features.

**Table 7.** Comparing the performance of different models with different learning methods.

| Model | Feature Learning | | Eng. Features | |
|---|---|---|---|---|
| | First Scenario | Second Scenario | First Scenario | Second Scenario |
| MLP | 75% | 70% | 79% | 74% |
| 1D-CNN | 94% | 90% | 82% | 76% |
| CNN-LSTM | 97% | 95% | 80% | 77% |
| 2D-CNN | **98%** | **96%** | 81% | 75% |

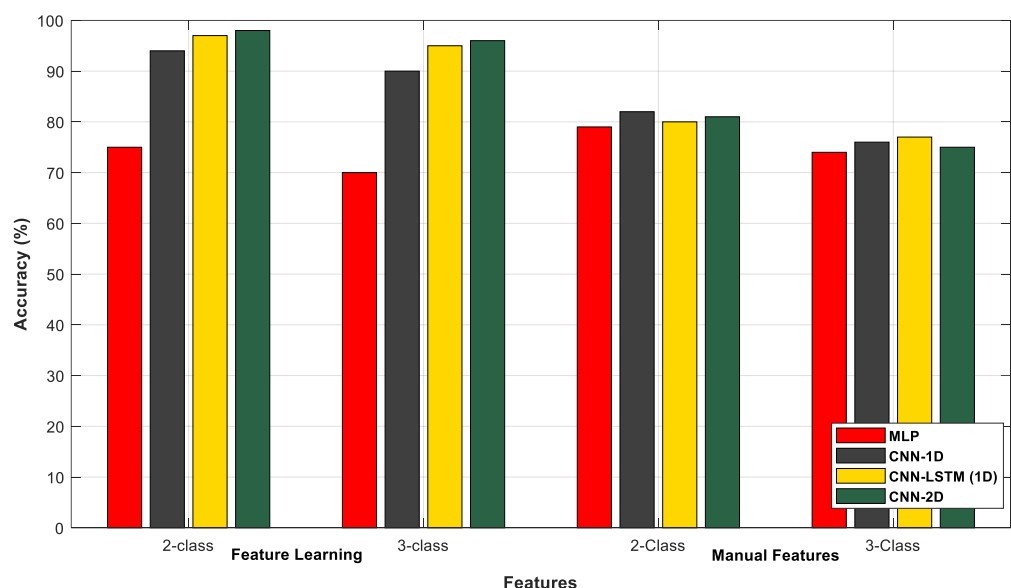

**Figure 13.** Bar diagram comparing different models with different learning methods.

This result is related to these networks' distinct architecture, which can automatically extract useful features from raw data for classification. Furthermore, obtaining engineering features necessitates expertise and prior knowledge, whereas learning features from raw data allows less specialized knowledge. While CNN-1D, CNN-LSTM (1D), and proposed CNN-2D deep networks perform better when learning features from raw data, all investigated models, including CNN-1D, CNN-LSTM (1D), CNN-2D, and MLP networks, performed nearly identically when learning engineering features. This demonstrates that deep networks cannot outperform traditional methods in emotion recognition without feature learning ability.

The nature of brain signals indicates that they have a low signal-to-noise ratio (SNR) and are highly sensitive to noise. This issue may make different classes' classification accuracy difficult. As a result, it is necessary to design the proposed network in order to classify different emotions in a way that is less sensitive to environmental noises. As a result, in this study, we artificially tested the performance of the proposed model in noisy environments. Gaussian white noise with different SNRs was added to the data for this purpose. Figure 14 depicts the classification results in noisy environments obtained using the proposed model. As is well known, the proposed deeply customized model has a high noise resistance when compared with other networks. This subject is related to personalized architecture (use of large filter dimensions in the initial layer of the network and use of tuned filter dimensions in the middle layers).

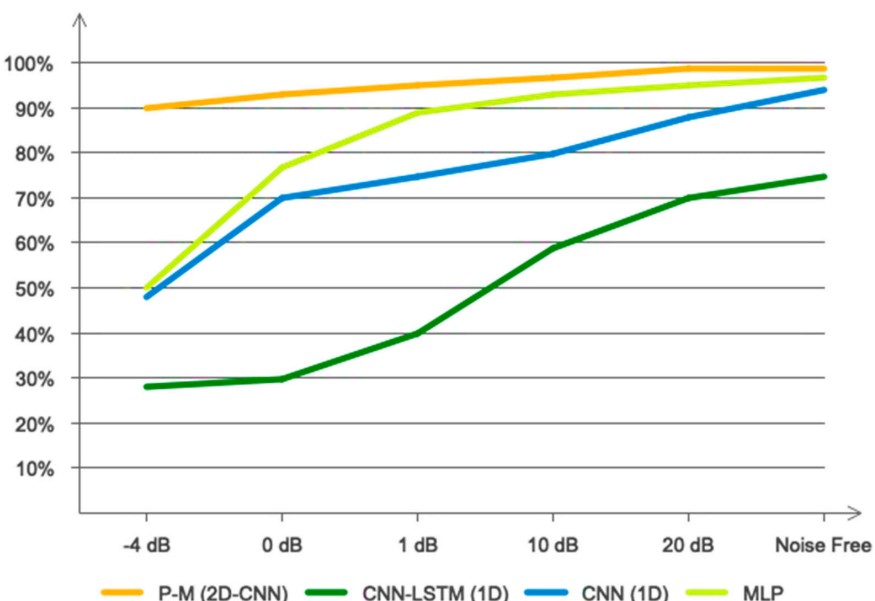

**Figure 14.** Bar diagram comparing different models with different learning methods.

Despite its positive results, this study, like all others, has benefits and drawbacks. One of this study's limitations is the small number of emotional classes. To that end, the number of emotional classes in the collected database should be increased. To address existing uncertainties, we plan to use a Generative Adversarial Network (GAN) instead of classical data augmentation and Type 2 Fuzzy Networks in conjunction with CNN in the future. The proposed architecture is suitable for real-time applications due to its simple and end-to-end architecture.

## 5. Discussion

In this section, the possible applications of the present study are reviewed along with the practical implications of the developed emotion recognition methodology for the new Society 5.0 paradigm.

The potential to provide machines with emotional intelligence in order to improve the intuitiveness, authenticity, and naturalness of the connection between humans and robots is an exciting problem in the field of human–robot interaction. A key component in doing this is the robot's capacity to deduce and comprehend human emotions. Emotions, as previously noted, are vital to the human experience and influence behavior. They are fundamental to communication, and effective relationships depend on having emotional intelligence or the capacity to recognize, control, and use one's emotions. The goal of affective computing is to provide robots with emotional intelligence to enhance regular human–machine interaction. (HMI). It is envisaged that BCI would enable robots to exhibit human-like observation, interpretation, and emotional expression skills. Following are the three primary perspectives that have been used to analyze emotions [40]:

a.  Formalization of the robot's internal emotional state: Adding emotional characteristics to agents and robots can increase their efficacy, adaptability, and plausibility. Determining neurocomputational models, formalizing them in already-existing cognitive architectures, modifying well-known cognitive models, or designing specialized emotional architectures has, thus, been the focus of robot design in recent years.

b.  Robotic emotional expression: In situations requiring complicated social interaction, such as assistive, educational, and social robotics, the capacity of robots to display recognisable emotional expressions has a significant influence on the social interaction that results.

c.  Robots' capacity to discern human emotional state: Interacting with people would be improved if robots could discern and comprehend human emotions.

According to the desired performance of the present study, the proposed model can be used in each of the discussed cases.

## 6. Conclusions

In this paper, a new model for automatic emotion recognition using EEG signals was developed. A standard database was collected for this purpose by music stimulation using EEG signals to recognize three classes of positive, negative, and neutral emotions. A Deep Learning model based on two-dimensional CNN networks was also customized for feature selection/extraction and classification operations. The proposed network, which included six convolutional layers and two fully connected layers, could classify three emotions in two different scenarios with 95% accuracy. Furthermore, the architecture suggested in this study was tested in a noisy environment and yielded acceptable results across a wide range of SNRs. As a result, even at $-4$ dB, the categorization accuracy of greater than 90% was maintained. In addition, the proposed method was compared with previous methods and studies in terms of different measuring criteria, and it had a promising performance. According to the favorable results of the proposed model, it can be used in real-time emotion recognition based on BCI systems.

**Author Contributions:** Conceptualization, F.B.; methodology, S.S. and S.D.; software, A.F. and F.B; validation, S.D. and S.S.; writing—original draft preparation, F.B. and A.F. All authors have read and agreed to the published version of the manuscript.

**Funding:** This research received no external funding.

**Data Availability Statement:** The data related to this article is publicly available on the GitHub platform under the title Baradaran emotion dataset.

**Conflicts of Interest:** The authors declare no conflict of interest.

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
