# Peer review of "Customized 2D CNN Model for the Automatic Emotion Recognition Based on EEG Signals"

_electronics, doi:10.3390/electronics12102232_

Round 1
Reviewer 1 Report
Brain-Computer Interface (BCI) systems employ various algorithms for the automatic classification of specific features in EEG signal sequences or patterns. Some frequent drawbacks of existing algorithms are their lack of stability, relatively high error rates, and poor accuracy are still considered as the central gap of this research. The study reported here was aimed at overcoming some of these. A Deep Convolutional Neural Network was conceive to classify (discriminate between) positive, negative and neutral emotional features from EEG signals recorded in response to positively or negatively experienced musical stimuli. Different feature learning algorithms were tested and compared with respect to their classification accuracy of positive, neutral, and negative emotional states. The results show classification accuracy ranging between 93% and 96%, which is promising compared with results from previous research using other classification models. When tested under noisy conditions, the classification accuracy of the neural network remains above 90%.
The article is reasonably well-written. The technical details are not always clear, some explanations are missing altogether; the conclusions need to be justified by further evidence-based argumentation.
“Methodology” has flaws and requires revisions, for example:
1) It is written that “to identify active channels, active electrodes were identified according to [28, 29], and the rest of the electrodes were excluded from the processing process” – this is not enough to understand what was done; the reader has to be able to understand the whole process here without having to check [28,29]. The essential information regarding criteria for inclusion/exclusion of electrodes needs to be provided in the text here.
2) Then, further on in the text: “segmentation and overlapping operations are performed on the data as follows: in the first scenario, first, for each electrode, 5 minutes (300 seconds) of the signal are selected for the positive state and the negative state; In this case, we will have 2 types of data with dimensions of 75000 (the sampling frequency is equal to 250) for each class, then the data of each channel is divided into 8-second intervals with the overlapping technique (to avoid the overfitting phenomenon). So, in fact, each electrode with a length of 75000 (300 seconds) is divided into 540 279 samples with a length of 2000 (8 seconds) with a time shift.” This text is not clear and needs to be rewritten: 8-second intervals for a given 300-second sequence? An electrode has no length, the signal sequence recorded from it does…..etc.
“Epoch” is misspelled in the manuscript as “Epock”, this must be corrected (Figure 4 and elsewhere in the text)
“Results” has minor flaws and requires revisions, for example
1) Figure 10 does not clearly show the differences in accuracy; scale the x-axis between 90%-100% to improve the visualization
2) Figure 12 is unnecessary under the light of the data given in Table 4. If you want to keep Figure 12, you must discuss the Receiver Operating Characteristics (ROC) curves in the text, and explain clearly what they mean
“Conclusions” are too minimalistic and require revisions
“Because of the proposed model's good performance, it can be used in BCI applications with high reliability” …? This needs to be justified better.
Author Response
Reviewer#1:
Comments:
Brain-Computer Interface (BCI) systems employ various algorithms for the automatic classification of specific features in EEG signal sequences or patterns. Some frequent drawbacks of existing algorithms are their lack of stability, relatively high error rates, and poor accuracy are still considered as the central gap of this research. The study reported here was aimed at overcoming some of these. A Deep Convolutional Neural Network was conceive to classify (discriminate between) positive, negative and neutral emotional features from EEG signals recorded in response to positively or negatively experienced musical stimuli. Different feature learning algorithms were tested and compared with respect to their classification accuracy of positive, neutral, and negative emotional states. The results show classification accuracy ranging between 93% and 96%, which is promising compared with results from previous research using other classification models. When tested under noisy conditions, the classification accuracy of the neural network remains above 90%.
The article is reasonably well-written. The technical details are not always clear, some explanations are missing altogether; the conclusions need to be justified by further evidence-based argumentation.
- ⎫ Thanks to the esteemed reviewer, we believe that your comments have been very useful and effective in enhancing the scientific and writing framework of the manuscript. We have considered all the comments in their entirety and made every effort to correct the manuscript in the manner suggested by the honorable reviewer.
- 1. Methodology” has flaws and requires revisions, for example:
- A) It is written that “to identify active channels, active electrodes were identified according to [28, 29], and the rest of the electrodes were excluded from the processing process” – this is not enough to understand what was done; the reader has to be able to understand the whole process here without having to check [28,29]. The essential information regarding criteria for inclusion/exclusion of electrodes needs to be provided in the text here.
- ⎫ The manuscript is revised based on this comment. According to the reviewer's opinion, the reason for choosing active electrodes for processing was explained more clearly.
“As it is clear, the use of all EEG channels will increase the computational load due to the increase in the dimensions of the feature matrix. Therefore, it is necessary to identify active channels. Emotion-related EEG electrodes have been mainly distributed in the pre-frontal lobe, temporal lobe margin, and posterior occipital lobe [35]. These regions are precisely in line with the physiological principle of emotion generation. By selecting the electrode distribution, the extracted feature dimension can be greatly reduced. The complexity of calculation can be diminished, and the experiment is simpler and easy to carry out. Based on what was said, only the electrodes of the prefrontal lobe, temporal lobe margin, and posterior occipital lobe, which include Pz, T3, C3, C4, T4, F7, F3, Fz, F4, F8, Fp1 and Fp2, have been used for processing [35].”
Which is highlighted in section 3.1, page 10, lines 325-334 and Ref [35].
- B) Then, further on in the text: “segmentation and overlapping operations are performed on the data as follows: in the first scenario, first, for each electrode, 5 minutes (300 seconds) of the signal are selected for the positive state and the negative state; In this case, we will have 2 types of data with dimensions of 75000 (the sampling frequency is equal to 250) for each class, then the dSata of each channel is divided into 8-second intervals with the overlapping technique (to avoid the overfitting phenomenon). So, in fact, each electrode with a length of 75000 (300 seconds) is divided into 540 279 samples with a length of 2000 (8 seconds) with a time shift.” This text is not clear and needs to be rewritten: 8-second intervals for a given 300-second sequence? An electrode has no length, the signal sequence recorded from it does…..etc.
- ⎫ The manuscript is revised based on this comment. According to the reviewer's opinion, section 3.1 was divided into smaller paragraphs to make it clearer for the readers, and this section was also rewritten and written briefly.
Which is highlighted in section 3.1, page 10, lines 324-346 and Ref [35].
- 2. “Epoch” is misspelled in the manuscript as “Epock”, this must be corrected (Figure 4 and elsewhere in the text).
- ⎫ The manuscript is revised based on this comment. Yes, the opinion of the honorable reviewer is absolutely correct. The word "Epoch" was corrected throughout the manuscript.
Which is highlighted in Fig 5.
- 3. “Results” has minor flaws and requires revisions, for example
- A) Figure 10 does not clearly show the differences in accuracy; scale the x-axis between 90%-100% to improve the visualization.
- ⎫ The manuscript is revised based on this comment. Yes, the opinion of the honorable reviewer is absolutely correct. According to the second reviewer, because the results of the bar chart were presented in Table 4, he requested to remove the bar chart. For this reason, this figure has been removed from the manuscript.
- B) Figure 12 is unnecessary under the light of the data given in Table 4. If you want to keep Figure 12, you must discuss the Receiver Operating Characteristics (ROC) curves in the text, and explain clearly what they mean
- ⎫ The manuscript is revised based on this comment. Based on the reviewer's opinion, a more complete description of the ROC curve was added to the manuscript.
“Figure 12 depicts the Receiver Operating Characteristic Curve (ROC) analysis for different scenario classifications. A ROC is a graphical plot that illustrates the diagnostic ability of a classifier system as its discrimination threshold is varied. In this diagram, the farther the curve is from the bisector and the closer it is to the vertical line on the left, the better the performance of the classifier. As is well known, each emotion class in the ROC analysis has a score between 0.9 and 1, indicating excellent classification performance.”
Which is highlighted in section 4, page 14, and lines 414-420.
- 4. “Conclusions” are too minimalistic and require revisions. “Because of the proposed model's good performance, it can be used in BCI applications with high reliability” …? This needs to be justified better.
- ⎫ The manuscript is revised based on this comment. According to the reviewer's opinion, the conclusion section of the manuscript was improved and revised.
“In this paper, a new model for automatic emotion recognition using EEG signals was developed. A standard database was collected for this purpose by music stimulation using EEG signals to recognize three classes of positive, negative, and neutral emotions. A deep learning model based on two-dimensional CNN networks was also customized for feature selection/extraction and classification operations. The proposed network, which included 6 convolutional layers and 2 fully connected layers, could classify three emotions in two different scenarios with a 95% accuracy. Furthermore, the architecture suggested in this study was tested in a noisy environment and yielded acceptable results across a wide range of SNRs. As a result, even at -4 dB, the categorization accuracy maintained above 90%. Also, the proposed method was compared with previous methods and studies in terms of different measuring criteria and it had a promising performance. According to the favorable results of the proposed model, it can be used in real-time emotion recognition based on BCI systems.”
Which is highlighted in Conclusions, page 19, and lines 541-547.

Reviewer 2 Report
This work presents a Customized 2D CNN Model for the Automatic Emotion Recognition based on EEG Signals. Emotion recognition based on EEG, have good acceptance at community, and the present work brings a general good explanation and solution for that. Although, the present paper didn't bringd huge innovation on that, but the study is interesting.
Good writing practices are recomended. For a good papaer is recommended avoid the use of temporal word, e.g., 'in recente time', 'recently', and so on. The present paper use the words 'recent studies' and 'recent years' 12 times along the paper. I strongly recommend the author to repalce it for a more not-temporal expressions.
L12: ".. and IT have achieved..."
L13: ".. gap of thESE researchES..."
L16: "... is presented AND IT can classify 3 positive,..."
L20: Please, re-weite the sentence. Replace 'bilayer' (not mfriendly word on the present context) for a more clear term.
Please. Add the questionnaire structure in the text or append it like append file. Was this questionnaire standardized and validated before on another research? Please, cite it! It is important to use standard questionnaires.
L94: Olher researches based on emotion and aviation, also use wavelets and other fatures for emotion recognition. based on EEG It is important to consider it.
In general, the INTRODUTION presents too LONG paragraphs. It is importart to organize it better, to improve the reading quality of the journal.
References 27 and 31 seem to have low significance in the text. There are several good researches since 2019 to 2022 and it is important to considere it better.
The papers didn't present the features used in the recognition process. With type of features? Statistical, Mean features (e.g. MARD, MAD), Wavelets, SFFT, ..Which one? It is important to express ther features used.
Figure 8: Validation Acc and Validation Loss seem to have no meaning in the plot, since one are presented how the complement of the other (1-p). Try to improve the plot. The same for train Acc and train Loss.
L259: The author said: '.., considering that emotional arousal occurs only between the ranges of 0.5 to 45 Hz, ..'.. Where is the source of this information? Where are the references of that? Please, add it.
L358: "Python programming environment.." IS NOT CORRECT expression. The IDE (environment) is irrelevant in the process. Maybe the author tryied to say, Python programming language.
Table 3: Reduce the width of the Table. Try to replace big expressions i.e. "Number of ...." and "Convolution Layers..", by another clear expression or abbreviation.
L361: Add [,] after to cite Figure and/or Table at beginning of a paragraph or sentence.
Figure 7: Caption of the Figure 7 is not OK. Please write something like: "Dataset distribution along the data recognition process", "Dataset distribution ..."
Figure 2, have too small words. Please, increase it. Too small words in a diagram, makes no sense to the readers and to the community.
Equation 1: Is rally poor of explanation. In general, output layers on ANN, CNN, DNN, uses 'Y', also the notation X to represent the input layers. Really complex notations to show important Equations, is not a good practice of papers. Please, improve the equation being more useful and clear to the readers. Other books brings better notation for the present work.
Add like append, the files of the EEG signals. It is important to see the shape of the eeg waves in this case. At least, the Beta-band.
The EEG bands were filteres and processed by the authors?
The paper use 'volunteer' and 'participant' to represent the same kind of people. Please uniform it in the text.
Section 3.1.: The author wrote only one huge paragraph. Please, wrote better the text and sometimes, avoid to use too long paragraphs; it makes the reading so boring.
Figure 4: Increase the size of the words. In addition, review all figures and increase for all if necessary.
Final remarks: The present paper have to many things to fix like, english expressions, wrinting structures, data nor presented like features used in more details, the EEG data (append), and so on. For this reason, I accept after to fix and the authors review all the text, mainly the references, the development steps and RESULTS. Sometimes the present paper seems to have too many information, but it aren't present clearly.
To conclude. The present work don't have good innovation. I mean, it is very common to find works like this, but the authors, need to show better, what is the good innovation they bring to us, to the readers. I can't see it in the text. Look to papers from journals like electronics, sensors, assistive, aerospace and the authors will find several works like the present here.
Best regards and good luck!
Author Response
Reviewer#2:
Comments:
This work presents a Customized 2D CNN Model for the Automatic Emotion Recognition based on EEG Signals. Emotion recognition based on EEG, have good acceptance at community, and the present work brings a general good explanation and solution for that. Although, the present paper didn't bringd huge innovation on that, but the study is interesting.
- ⎫ Thanks to the esteemed reviewer, we believe that your comments have been very useful and effective in enhancing the scientific and writing framework of the manuscript. We have considered all the comments in their entirety and made every effort to correct the manuscript in the manner suggested by the honorable reviewer.
The contribution of this study is organized as follows:
- a. Collecting a comprehensive database of emotion recognition using musical stimulation based on EEG signals.
- b. Presenting an intelligent model based on deep learning in order to separate two and three basic emotional classes.
- c. Using end-to-end deep neural networks, which has led to the elimination of feature selection/extraction block diagram
- d. Providing an algorithm based on deep convolutional networks that can be resistant to environmental noise to an acceptable extent.
- e. Presenting an automatic model that can classify two and three emotional classes with the highest accuracy and the least error compared to previous research.
Which is highlighted in section 1, page 4, and lines 158-168.
- 1. Good writing practices are recomended. For a good papaer is recommended avoid the use of temporal word, e.g., 'in recente time', 'recently', and so on. The present paper use the words 'recent studies' and 'recent years' 12 times along the paper. I strongly recommend the author to repalce it for a more not-temporal expressions.
- ⎫ The manuscript is revised based on this comment. Yes, the opinion of the honorable reviewer is absolutely correct. The requested items were corrected and highlighted in the whole manuscript.
- A) L12: ".. and IT have achieved..."
- ⎫ The manuscript is revised based on this comment.
- B) L13: ".. gap of thESE researchES..."
- ⎫ The manuscript is revised based on this comment.
- C) L16: "... is presented AND IT can classify 3 positive,..."
- ⎫ The manuscript is revised based on this comment.
- D) L20: Please, re-weite the sentence. Replace 'bilayer' (not mfriendly word on the present context) for a more clear term.
- ⎫ The manuscript is revised based on this comment. According to the opinion of the respected reviewer, the mentioned sentence was rewritten and modified.
- 2. Add the questionnaire structure in the text or append it like append file. Was this questionnaire standardized and validated before on another research? Please, cite it! It is important to use standard questionnaires.
- ⎫ The manuscript is revised based on this comment. Yes, the questionnaire used is a standard questionnaire that is used in almost all research related to the automatic detection of emotions [22, 28]. According to the opinion of the respected reviewer, the explanations related to the used questionnaire were added to the manuscript along with the reference. Also, the questionnaire used in the side files of the manuscript was attached for the journal admin.
“SAM is a brief self-report questionnaire that uses images to express the scales of pleasure, arousal, and dominance. Given its non-verbal design, the questionnaire is readable to people regardless of age, language skill or other educational factors [15].“
“Before the experiment, Beck's popular Depression Mood Questionnaire [29] was used to exclude depressed volunteers from the experiment. After this test, the candidates who received a score higher than 21, have depression disorder and were excluded from the test process. The reason for using this test is that depressive disorder causes a lack of emotional induction in people. In addition, the SAM assessment test in paper form and 9 grades has been used to control the dimension of valence and arousal [15].“
Which is highlighted in section 1 and 2, pages 2 and 4, lines 58-60 and lines 189-195.
- 3. L94: Olher researches based on emotion and aviation, also use wavelets and other fatures for emotion recognition. based on EEG It is important to consider it.
- ⎫ With respect to the opinion of the esteemed reviewer, we have reviewed about 10 recent researches and presented the advantages and disadvantages of each one. However, based on the opinion of the respected reviewer, we have added and reviewed 3 studies in 2023 based on wavelet transform and deep learning to detect emotions from EEG signals in the review section of previous studies.
Which is highlighted in section 1, page 3, and lines 112-128.
- 4. In general, the INTRODUTION presents too LONG paragraphs. It is importart to organize it better, to improve the reading quality of the journal.
- ⎫ The manuscript is revised based on this comment. Respecting the opinion of the respected reviewer, unnecessary material was removed from the introduction section.
- 5. References 27 and 31 seem to have low significance in the text. There are several good researches since 2019 to 2022 and it is important to considere it better.
- ⎫ The manuscript is revised based on this comment. According to the opinion of the respected referee, references 33 and 37 have been replaced with newer references related to 2023.
- 6. The papers didn't present the features used in the recognition process. With type of features? Statistical, Mean features (e.g. MARD, MAD), Wavelets, SFFT, ..Which one? It is important to express ther features used.
- ⎫ With respect to the opinion of the honorable reviewer, as explained in manuscript (Section 3.1), we have used the 8th Daubechies WT to extract 3 frequency bands of alpha, beta and gamma. Also, an example of the extracted bands is shown in Figure 4 for the first subject. In the next step, the 3 frequency bands extracted as images are entered into the customized deep network architecture.
- ⎫ DNN is generally considered as a black box. The inner working mechanism of DNN is difficult to understand and none of the previous studies have attempted to explore it. At the most basic level, "black box" just means that for deep neural networks, we don't know how all the individual neurons work together to get to the final output. Many times it is not even clear what a particular neuron does alone and what deep features are extracted from the data. Also, the type of deep features extracted in deep learning has not been previously discussed in any other study [1-5].
[1] W.S. McCulloch & W. Pitts, “A logical calculus of the ideas immanent in nervous activity”. Bulletin of Mathematical Biophysics 5, 115–133 (1943).
[2] Borjali, A., Chen, A. F., Muratoglu, O. K., Morid, M. A., & Varadarajan, K. M. (2020). Deep learning in orthopedics: How do we build trust in the machine?. Healthcare Transformation.
[3] P.J. Blazek & M.M. Lin. “Explainable neural networks that simulate reasoning”. Nature Computational Science 1, 607–618 (2021).
[4] Zhang, Z., Sugino, T., Akiduki, T., & Mashimo, T. (2019, July). A study on development of wavelet deep learning. In 2019 International Conference on Wavelet Analysis and Pattern Recognition (ICWAPR) (pp. 1-6). IEEE.
[5] P.J. Blazek. “How Aristotle is fixing deep learning’s flaws”. The Gradient (2022).
- 7. Figure 8: Validation Acc and Validation Loss seem to have no meaning in the plot, since one are presented how the complement of the other (1-p). Try to improve the plot. The same for train Acc and train Loss.
- ⎫ With respect to the opinion of the respected reviewer, Figure 9 shows the process of training and testing the model in order to classify emotions in 200 repetitions. The reason for reporting the accuracy and error of training and validation of the model is to show that overfitting did not occur during the training and testing process. Also, it indicates that the proposed model has converged to the final value in several iterations. In addition to this error, training and validation have been provided to show whether the network error has decreased or increased by increasing the repetition of the algorithm.
- ⎫ This fig has already been reported in other references [1-4] in the same way and title.
[1] Shahini, N., Bahrami, Z., Sheykhivand, S., Marandi, S., Danishvar, M., Danishvar, S., & Roosta, Y. (2022). Automatically Identified EEG Signals of Movement Intention Based on CNN Network (End-To-End). Electronics, 11(20), 3297. Figs 7-10
[2] Khaleghi, N., Rezaii, T. Y., Beheshti, S., Meshgini, S., Sheykhivand, S., & Danishvar, S. (2022). Visual Saliency and Image Reconstruction from EEG Signals via an Effective Geometric Deep Network-Based Generative Adversarial Network. Electronics, 11(21), 3637. Fig 11
[3] Sheykhivand, S., Mousavi, Z., Mojtahedi, S., Rezaii, T. Y., Farzamnia, A., Meshgini, S., & Saad, I. (2021). Developing an efficient deep neural network for automatic detection of COVID-19 using chest X-ray images. Alexandria Engineering Journal, 60(3), 2885-2903. Figs 9-10
[4] Mousavi, Z., Shahini, N., Sheykhivand, S., Mojtahedi, S., & Arshadi, A. (2022). COVID-19 detection using chest X-ray images based on a developed deep neural network. SLAS technology, 27(1), 63-75. Figs 4
- 8. L259: The author said: '.., considering that emotional arousal occurs only between the ranges of 0.5 to 45 Hz, ..'.. Where is the source of this information? Where are the references of that? Please, add it.
- ⎫ The manuscript is revised based on this comment. With respect to the reviewer's opinion, the reference related to the frequency range useful for the automatic detection of emotions has been added to the manuscript.
Which is highlighted in section 3.1, and Ref [22, 34].
- 9. L358: "Python programming environment.." IS NOT CORRECT expression. The IDE (environment) is irrelevant in the process. Maybe the author tryied to say, Python programming language.
- ⎫ The manuscript is revised based on this comment. Thanks for the accuracy of the opinion of the honorable reviewer, yes, we mean the Python programming language. However, in other references (See the figure below) they have also used this phrase. Respecting the opinion of the honorable reviewer, the relevant phrase was modified in the manuscript.
Which is highlighted in section 4, lines 396-398.
- 10. Table 3: Reduce the width of the Table. Try to replace big expressions i.e. "Number of ...." and "Convolution Layers..", by another clear expression or abbreviation.
- ⎫ The manuscript is revised based on this comment. Table 3 was modified and improved according to the opinion of the respected reviewer.
Which is highlighted in Table 3.
- 11. L361: Add [,] after to cite Figure and/or Table at beginning of a paragraph or sentence.
- ⎫ The manuscript is revised based on this comment.
- 12. Figure 7: Caption of the Figure 7 is not OK. Please write something like: "Dataset distribution along the data recognition process", "Dataset distribution ..."
- ⎫ The manuscript is revised based on this comment. According to the reviewer's opinion, the caption related to Figure 7 was changed to the requested form.
Which is highlighted in Figure 8 (fig 7), and page 12.
- 13. Figure 2, have too small words. Please, increase it. Too small words in a diagram, makes no sense to the readers and to the community.
- ⎫ The manuscript is revised based on this comment. According to the reviewer's opinion, the text related to Figure 3 (Fig 2) was enlarged and the figure was modified.
Which is highlighted in Figure 3, and page 9.
- 14. Equation 1: Is rally poor of explanation. In general, output layers on ANN, CNN, DNN, uses 'Y', also the notation X to represent the input layers. Really complex notations to show important Equations, is not a good practice of papers. Please, improve the equation being more useful and clear to the readers. Other books brings better notation for the present work.
- ⎫ The manuscript is revised based on this comment. According to the reviewer's opinion, equation 1 was removed and according to the deep learning book published by Nature in 2023 [33], a more comprehensible correct equation was replaced.
Which is highlighted in section 2.2.2., page 8 and lines 279-288.
- 15. Add like append, the files of the EEG signals. It is important to see the shape of the EEG waves in this case. At least, the Beta-band.
- ⎫ The manuscript is revised based on this comment. According to the opinion of the respected reviewer, an example of the recorded signal for all three emotional classes, positive, negative and neutral, from 2 electrodes F4 and C4 is presented in Figure 1.
“An example of the recorded EEG signal for 3 different emotional states is shown in Figure 1.
Figure 1. An example of EEG signal recorded from C4 and F4 channels for positive, negative and neutral emotions in subject 1.
- ⎫ Obviously, after the paper acceptance, the dataset registered on the GitHub will be freely accessible for researchers to use.
- ⎫ The ethics committee of our university emphasizes that in order to avoid conflict of interest and biases and other policies, the registered EEG files should be made publicly available after the article has been approved.
Which is highlighted in Figure 1, and lines 216-221.
- 16. The EEG bands were filteres and processed by the authors?
- ⎫ Yes, all stages of signal collection, signal preprocessing and architectural design have been done by the authors of the article, the details of filtering and preprocessing are explained in sections 3.1 and 3.2.
- 17. The paper use 'volunteer' and 'participant' to represent the same kind of people. Please uniform it in the text.
- ⎫ The manuscript is revised based on this comment. According to the reviewer's opinion, the requested word has been unified throughout the manuscript.
- 18. Section 3.1.: The author wrote only one huge paragraph. Please, wrote better the text and sometimes, avoid to use too long paragraphs; it makes the reading so boring.
- ⎫ The manuscript is revised based on this comment According to the reviewer's opinion, section 3.1 was divided into smaller paragraphs to make it clearer for the readers, and this section was also rewritten and written briefly.
Which is highlighted in page 10, section 3.1 and lines 314-345.
- 19. Figure 4: Increase the size of the words. In addition, review all figures and increase for all if necessary.
- ⎫ The manuscript is revised based on this comment. According to the reviewer's opinion, the text related to Figure 5 (fig 4) was enlarged and the figure was modified.
Which is highlighted in page 11 and Fig 5.
- 20. Final remarks: The present paper have to many things to fix like, english expressions, wrinting structures, data nor presented like features used in more details, the EEG data (append), and so on. For this reason, I accept after to fix and the authors review all the text, mainly the references, the development steps and RESULTS. Sometimes the present paper seems to have too many information, but it aren't present clearly.
To conclude. The present work don't have good innovation. I mean, it is very common to find works like this, but the authors, need to show better, what is the good innovation they bring to us, to the readers. I can't see it in the text. Look to papers from journals like electronics, sensors, assistive, aerospace and the authors will find several works like the present here.
- ⎫ While respecting the respected reviewer, we would like to thank you for your careful opinion on the present manuscript and we believe that your comments have greatly helped to improve the manuscript. We have considered almost all the points raised by you in the manuscript. The innovations of the present study, which were mentioned in the first comment, have been examined in detail. Also, a lot of effort has been made to collect data; So that, the process of recording brain signals from the participants and obtaining the necessary permits has lasted for 6 months. We hope that the manuscript after possible acceptance can fix the shortcomings of previous studies and the collected database can be used and evaluated by other researchers in this field.

Reviewer 3 Report
This article describes the development of a new model for automatic emotion recognition using EEG signals, based on Deep Convolutional Neural Networks. This is a nice work. Overall, the article is written in good English.
In my opinion some improvements can be made:
- The hardware/software used should be described in more detail.
- The data collection and mathematical filtering methods should be included.
- A bullet point is missing in line 148.
- The age of the human participants in the experiments is always near 24. It is also interesting to test and validate the proposed model with younger and older human participants.
- How were data from previous studies obtained and tested, to validate the comparison with the proposed model?
- The conclusions should be extended, to reflect the results obtained.
- The authors should also mention some possible applications for this work, e.g. what are the practical implications of the developed emotion recognition methodology for the new paradigm of Society 5.0.
Author Response
Reviewer#3:
Comments:
This article describes the development of a new model for automatic emotion recognition using EEG signals, based on Deep Convolutional Neural Networks. This is a nice work. Overall, the article is written in good English.
- ⎫ Thanks to the esteemed reviewer, we believe that your comments have been very useful and effective in enhancing the scientific and writing framework of the manuscript. We have considered all the comments in their entirety and made every effort to correct the manuscript in the manner suggested by the honorable reviewer.
- 1. The hardware/software used should be described in more detail.
- ⎫ The manuscript is revised based on this comment. According to the reviewer's opinion, the hardware and software used have been added to the manuscript with more details.
- ⎫ “In order to record EEG signals, a 21-channel Medicom device according to the 10-20 standard has been used. Medicam is a Russian device that is widely used in medical clin-ics and research centers for recording brain signals due to its high performance. Silver chloride electrodes, which were organized in the form of a hat, were used in this work. Two electrodes A1 and A2 have been used to reference brain signals. Thus, out of 21 channels, 19 channels are actually available. To avoid EOG signal artifacts, volunteers were asked to keep their eyes closed during EEG signal recording. The sampling frequency is 250 Hz and an impedance matching of 10 kΩ is used on the electrodes. The recording mode of the signals is also set as bipolar.”
- ⎫ “Python programming language under Keras and TensorFlow has been used to simulate the proposed deep model. All simulation results are extracted from a computer sys-tem with 16 GB RAM, a 2.8 GHz CPU, and a 2040 GPU.”
Which is highlighted in sections 2.1 and 4, pages 4 and 13 and lines 196-204 and 396-398.
- 2. The data collection and mathematical filtering methods should be included.
- ⎫ The manuscript is revised based on this comment. According to the reviewer's opinion, the filtering algorithms used in this study have been added to the manuscript and reviewed.
“In the field of signal processing, many filtering algorithms are used for signal pre-processing in order to remove motion and environmental noises and also to reach the desired frequency range. Among the most popular filters, we can mention Notch [30] and Butterworth filters [31], which are also used in this study. In the following, the mathematical details of each of these filters will be examined.
2.2.1. Notch filter
A Notch filter is a type of band-stop filter, which is a filter that attenuates frequencies within a specific range while passing all other frequencies unaltered. For a Notch filter, this range of frequencies is very narrow. The range of frequencies that a band-stop filter attenuates is called the stopband. The narrow stopband in a Notch filter makes the frequency response resemble a deep Notch, which gives the filter its name. It also means that notch filters have a high Q factor, which is the ratio of center frequency to bandwidth. Notch filters are used to remove a single frequency or a narrow band of frequencies. In audio systems, a Notch filter can be used to remove interfering frequencies such as power line hum. Notch filters can also be used to remove a specific interfering frequency in radio receivers and software-defined radio. The main application of the notch filter can be considered to remove the frequency of 50 or 60 Hz of city electricity [30].
2.2.2. Butterworth filter
Among the very popular filters for removing the frequency range above 100 Hz of brain signals, the low-pass Butterworth filter is [31]. The Butterworth filter is a type of signal processing filter designed to have a frequency response that is as flat as possible in the passband. It is also referred to as a maximally flat magnitude filter. One of the most important features of this filter is the existence of a flat maximum frequency response in the pass region and no ripple. Also, its graph tends to a very good approximation with a negative slope to negative infinity.
Butterworth showed that a low-pass filter could be designed whose cutoff frequency was normalized to 1 radian per second and its frequency response can be defined as the Eq (1):
|
(1) |
|
whereis the angular frequency in radians per second and n is the number of poles in the filter—equal to the number of reactive elements in a passive filter. If= 1, the amplitude response of this type of filter in the passband is 1/√2 ≈ 0.7071, which is half power or −3 dB [31]”.
Which is highlighted in section 2.2, page 7 and lines 233-267.
- 3. A bullet point is missing in line 148.
- ⎫ Thanks to the careful opinion of the respected reviewer, the manuscript was revised based on the mentioned matter.
- 4. The age of the human participants in the experiments is always near 24. It is also interesting to test and validate the proposed model with younger and older human participants.
- ⎫ Yes, the opinion of the honorable reviewer seems interesting. However, there are limitations. The average age of the participants is considered based on the recommendation in the study [22, 34 and 35]. According to this study, the most emotional induction occurs at this age. Also, unfortunately, the ethics committee of the university does not allow the use of people under the age of 18 and over the age of 50 in experiments due to its own policies.
- 5. How were data from previous studies obtained and tested, to validate the comparison with the proposed model?
- ⎫ The manuscript is revised based on this comment. Unfortunately, there is no benchmark EEG database for inducing emotions based on musical stimulation. This is the reason why in this research we have decided to collect an available EEG database for inducing emotions so that other researchers can easily use it in their studies.
Due to the fact that the public EEG database for inducing emotions based on musical stimulation was not available, for this reason we have implemented the popular and common methods and algorithms of recent researches with the collected database.
In this regard, two methods based on raw signal feature learning and engineering feature extraction (manual) will be used, along with MLP classifiers, CNN-1D, CNN-LSTM (1D), and the proposed CNN-2D model. The gamma band is extracted from the recorded EEG signals for engineering features (using a 5-level Daubechies WT). From the obtained gamma band, two Root Mean Square (RMS) and Standard Deviation (SD) features will be extracted. The input dimensions for the first and second scenarios will be and, respectively, based on this. Following that, MLP, CNN-1D, CNN-LSTM (1D), and proposed CNN-2D networks are used to classify the extracted feature vector. The raw signals are classified using expressed networks for feature learning, with no manual feature extraction or selection. The MLP network has two fully connected layers, the last of which has two neurons (for the two-class state) and three neurons (for the three-class state). According to [22], CNN-1D and CNN-LSTM (1D) network architectures are considered. To improve the performance of the expressed networks, their hyperparameters are adjusted based on the type of data. The results of this comparison are shown in Table 6 and Figure 13. According to Table 6, feature learning from raw signal for CNN-1D, CNN-LSTM (1D), and proposed CNN-2D deep networks has been continuously improved, and these networks can learn important features layer by layer, and 2 class and 3 class scenarios with accuracy greater than 90%. On the contrary, as can be seen from the engineering features used as input in CNN-1D, CNN-LSTM (1D), and CNN-2D deep networks, these networks have not improved in recognition. When feature learning and engineering features are compared, feature learning from raw data with CNN-1D, CNN-LSTM (1D), and CNN-2D deep networks outperforms engineering features. This result is related to these networks' distinct architecture, which can automatically extract useful features from raw data for classification. Furthermore, obtaining engineering features necessitates expertise and prior knowledge, whereas learning features from raw data necessitates less specialized knowledge. While CNN-1D, CNN-LSTM (1D), and proposed CNN-2D deep networks perform better when learning features from raw data, all investigated models, including CNN-1D, CNN-LSTM (1D), CNN-2D, and MLP networks, perform nearly identically when learning engineering features. This demonstrates that deep networks cannot outperform traditional methods in emotion recognition without feature learning ability.
The nature of brain signals indicates that they have a low signal-to-noise ratio (SNR) and are highly sensitive to noise. This issue may make different classes' classification accuracy difficult. As a result, it is necessary to design the proposed network in order to classify different emotions in a way that is less sensitive to environmental noises. As a result, in this study, we artificially tested the performance of the proposed model in noisy environments. Gaussian white noise with different SNRs was added to the data for this purpose. Figure 14 depicts the classification results in noisy environments obtained using the proposed model. As is well known, the proposed deeply customized model has a high noise resistance when compared to other networks. This subject is related to personalized architecture (use of large filter dimensions in the initial layer of the network and use of tuned filter dimensions in the middle layers).
Eng Features |
Feature Learning |
Model |
||
Second Scenario |
First Scenario |
Second Scenario |
First scenario |
|
74% |
79% |
70% |
75% |
MLP |
76% |
82% |
90% |
94% |
1D-CNN |
77% |
80% |
95% |
97% |
CNN-LSTM |
75% |
81% |
96% |
98% |
2D-CNN |
Table 6. Comparing the performance of different models with different learning methods.
Figure 13. Bar diagram comparing different models with different learning methods.
Which is highlighted in section 4, pages 15-17 and lines 444-480.
- 6. The conclusions should be extended, to reflect the results obtained.
- ⎫ The manuscript is revised based on this comment. According to the reviewer's opinion, the conclusion section of the manuscript was improved and revised.
“In this paper, a new model for automatic emotion recognition using EEG signals was developed. A standard database was collected for this purpose by music stimulation using EEG signals to recognize three classes of positive, negative, and neutral emotions. A deep learning model based on two-dimensional CNN networks was also customized for feature selection/extraction and classification operations. The proposed network, which included 6 convolutional layers and 2 fully connected layers, could classify three emotions in two different scenarios with a 95% accuracy. Furthermore, the architecture suggested in this study was tested in a noisy environment and yielded acceptable results across a wide range of SNRs. As a result, even at -4 dB, the categorization accuracy maintained above 90%. Also, the proposed method was compared with previous methods and studies in terms of different measuring criteria and it had a promising performance. According to the favorable results of the proposed model, it can be used in real-time emotion recognition based on BCI systems.”
Which is highlighted in conclusions section, page 19, and lines 534-546 [27].
- 7. The authors should also mention some possible applications for this work, e.g. what are the practical implications of the developed emotion recognition methodology for the new paradigm of Society 5.0.
- ⎫ The manuscript is revised based on this comment. According to the reviewer's opinion, a special section titled discussion regarding possible applications was considered for the present research.
- ⎫ “In this section, the possible applications of the present study are reviewed along with the practical implications of the developed emotion recognition methodology for the new Society 5.0 paradigm.
The potential to provide machines emotional intelligence in order to improve the intuitiveness, authenticity, and naturalness of the connection between humans and robots is an exciting problem in the field of human-robot interaction. A key component in doing this is the robot's capacity to deduce and comprehend human emotions. Emotions, as previously noted, are vital to the human experience and influence behavior. They are fundamental to communication, and effective relationships depend on having emotional intelligence, or the capacity to recognize, control, and use one's emotions. The goal of affective computing is to provide robots emotional intelligence to enhance regular human-machine interaction. (HMI). It is envisaged that BCI would enable robots to exhibit human-like observation, interpretation, and emotional expression skills. Following are the three primary perspectives that have been used to analyze emotions [40]:
Formalization of the robot's internal emotional state: Adding emotional characteristics to agents and robots can increase their efficacy, adaptability, and plausibility. Determining neurocomputational models, formalizing them in already-existing cognitive architectures, modifying well-known cognitive models, or designing specialized emotional architectures has thus been the focus of robot design in recent years;
Robotic emotional expression: In situations requiring complicated social interaction, such as assistive, educational, and social robotics, the capacity of robots to display recognisable emotional expressions has a significant influence on the social interaction that results.
Robots' capacity to discern human emotional state: Interacting with people would be improved if robots could discern and comprehend human emotions.
According to the desired performance of the present study, the proposed model can be used in each of the discussed cases.
Which is highlighted in section 6, page 18, and lines 504-546.
We are especially grateful to the respected reviewer for his time to review the present study.
With respect

Round 2
Reviewer 1 Report
The manuscript can now be accepted for publication. The language requires final editing by the English editors of MDPI, for example:
line 12: "has achieved" should replace "have achieved"
line 13: "this research" should replace "these researches"...
....the rest of the manuscript text needs also needs a final Editor's check for spelling and grammar.
Reviewer 3 Report
I appreciate the authors' efforts in addressing the comments and making necessary revisions to the article. The changes made have improved the overall clarity of the content, making it more informative and useful for the readers. I believe the article is now ready for publication.